# Going Deeper into Locally Differentially Private Graph Neural Networks

Longzhu He [1]   Chaozhuo Li [1]   Peng Tang [2]   Sen Su [1]

## Abstract

Graph Neural Networks (GNNs) have demonstrated superior performance in a variety of graph mining and learning tasks. However, when node representations involve sensitive personal information or variables related to individuals, learning from graph data can raise significant privacy concerns. Although recent studies have explored local differential privacy (LDP) to address these concerns, they often introduce significant distortions to graph data, severely degrading private learning utility (*e.g.*, node classification accuracy). In this paper, we present UPGNET, an LDP-based privacy-preserving graph learning framework that enhances utility while protecting user data privacy. Specifically, we propose a three-stage pipeline that generalizes the LDP protocols for node features, targeting privacy-sensitive scenarios. Our analysis identifies two key factors that affect the utility of privacy-preserving graph learning: *feature dimension* and *neighborhood size*. Based on the above analysis, UPGNET enhances utility by introducing two core layers: *High-Order Aggregator (HOA) layer* and the *Node Feature Regularization (NFR) layer*. Extensive experiments on real-world datasets indicate that UPGNET significantly outperforms existing methods in terms of both privacy protection and learning utility.

## 1. Introduction

In recent years, Graph Neural Networks (GNNs) have shown superior performance in various domains, including social sciences (Hamilton et al., 2017), graph mining (Li et al., 2019), and bioinformatics (Fout et al., 2017). GNNs have also achieved state-of-the-art performance on a range of downstream graph learning tasks, such as node classification (Kipf & Welling, 2017), link prediction (Zhang & Chen,

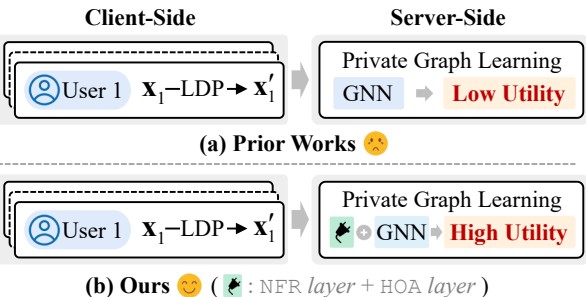

*Figure 1.* Comparison of **(a) prior works** and **(b) ours** in the locally private graph learning scenario. The scenario comprises a cloud server and multiple users situated across different clients. Users' sensitive node features $\mathbf{x}$ are perturbed to $\mathbf{x}'$ using LDP before uploading to the cloud server for graph learning. Our approach achieves higher utility by integrating than prior works.

2018), and community detection (Chen et al., 2019). In real-world scenarios, graphs frequently contain significant amounts of sensitive personal information, such as user profiles on social networks. However, recent studies (Wang & Wang, 2022; Shen et al., 2022; Zhang et al., 2022; Meng et al., 2023) have proposed various privacy attack methods targeting GNN models, which pose serious security and privacy challenges during their training. Therefore, designing an efficient privacy-preserving GNN framework to protect users' private information is of paramount importance.

To collect and analyze private data from decentralized data owners, local differential privacy (LDP) (Kasiviswanathan et al., 2011) has been increasingly accepted as the *de facto* standard for data privacy in the research community (Erlingsson et al., 2014; Ding et al., 2017; Wang et al., 2019a;b). In the LDP protocol, multiple users must interact with an untrustworthy server that may exploit their private data. Each user perturbs their data locally, typically through noise injection (Dwork et al., 2006), to ensure privacy. The perturbed data is then transmitted to the server, which conducts data analysis and learning based on this information. Locally private graph learning have recently gained significant attention from researchers (Sajadmanesh & Gatica-Perez, 2021; Lin et al., 2022; Jin & Chen, 2022; Pei et al., 2023). Fig. 1 illustrates this scenario, where a cloud server interacts with decentralized users. Each user perturbs their sensitive node features $\mathbf{x}$ to $\mathbf{x}'$ under LDP before submitting them to the server for private graph learning (*e.g.*, node classification).

[1] Beijing University of Posts and Telecommunication, Beijing, China [2] Shandong University, Qingdao, China. Correspondence to: Sen Su <susen@bupt.edu.cn>.

*Proceedings of the $42^{nd}$ International Conference on Machine Learning*, Vancouver, Canada. PMLR 267, 2025. Copyright 2025 by the author(s).

However, in locally private graph learning, existing LDP protocols for perturbing node features cause significant performance degradation in GNNs (Sajadmanesh & Gatica-Perez, 2021; Lin et al., 2022). Specifically, LDP regulates noise injection through the *privacy budget* $\epsilon$, where a smaller $\epsilon$ introduces more noise. In practice, $\epsilon$ is highly limited and node features are typically multidimensional with multiple attributes. As a result, each attribute receives only a minimal fraction of the budget, leading to substantial information loss. While server-side aggregation mitigates noise to some extent, it also introduces excessive estimation errors, further reducing utility (*e.g.*, node classification accuracy). Thus, the key challenge is *how to maximize the utility of privacy-preserving graph learning while ensuring user privacy*.

**Contributions.** To address this challenge, we present a three-stage pipeline that generalizes the current LDP protocols for perturbing node features. Our analysis of the pipeline reveals two key factors that influence the estimation error in feature aggregation: *feature dimension* and *neighborhood size*. We conclude that reducing the effective feature dimension and expanding the effective neighborhood size help minimize the estimation error and thus enhance the utility.

Based on these findings, we propose UPGNET, a utility-enhanced framework for locally privacy-preserving graph learning that minimizes estimation error and maximizes utility from two key perspectives (Comparison with prior works is presented in Fig. 1). First, to reduce the effective feature dimensions, we introduce a *Node Feature Regularization (NFR) layer* based on $L_1$-regularization (Bühlmann & Van De Geer, 2011), a classical optimization technique that promotes sparsity in solutions. We conduct a theoretical analysis using proximal gradient descent (Nitanda, 2014; Li & Lin, 2015) to derive a one-off, non-iterative solution. This enables the NFR to obtain sparse embeddings through feature selection, thereby reducing effective feature dimensions during aggregation. Second, to expand the effective neighborhood size, we propose a multi-hop aggregation method, the *High-Order Aggregator (HOA) layer*. Although our theoretical analysis indicates that increasing neighborhood size improves utility, real-world graphs often feature small neighborhoods. A straightforward solution, such as aggregating multi-layer node features (Abu-El-Haija et al., 2019; Gasteiger et al., 2019; Chen et al., 2020), suffers from over-smoothing (Rusch et al., 2023). As the number of layers increases, node embeddings tend to converge, diminishing the effectiveness of higher-order neighbors in correcting errors. To address this, our HOA layer leverages personalized aggregation and Dirichlet energy (Zhou et al., 2021; Rusch et al., 2023) analysis, effectively mitigating oversmoothing and reducing noise bias injection. UPGNET comprises two architectures: H-N (HOA followed by NFR) and N-H (NFR followed by HOA), both of which can be independently integrated with any GNN architecture. We

evaluate overall performance of UPGNET and the contributions of each component under different parameters through theoretical analysis and extensive experiments.

Our contributions are summarized as follows. ① We propose a three-stage pipeline to systematically generalize the LDP protocols for perturbing node features. By analyzing the pipeline, we identify two key factors influencing the estimation error of feature aggregation. ② Based on the above analysis, we propose NFR and HOA layers to reduce the estimation error and integrate them with LDP protocols, introducing UPGNET, a utility-enhanced privacy-preserving graph learning framework. ③ Extensive experiments on real datasets demonstrate that UPGNET excels in achieving privacy preservation and superior graph learning utility.

## 2. Preliminaries

This section first defines the problem (Sec. 2.1), then provides the essential background on LDP (Sec. 2.2) and GNN (Sec. 2.3), and finally describes the threat model (Sec. 2.4).

### 2.1. Problem Definition

Consider a graph $\mathcal{G} = (\mathcal{V}, \mathcal{E})$, where $\mathcal{V} = \{v_1, v_2, \ldots, v_{|\mathcal{V}|}\}$ represents the set of nodes and $\mathcal{E}$ is the set of edges. Each decentralized user $v \in \mathcal{V}$ locally possesses a $d$-dimensional feature vector $\mathbf{x}_v \in \mathbb{R}^d$, and the feature matrix is defined as $\mathbf{X} \in \mathbb{R}^{|\mathcal{V}| \times d}$. Following previous work (Sajadmanesh & Gatica-Perez, 2021; Lin et al., 2022; Pei et al., 2023), we assume that the server has access to $\mathcal{V}$ and $\mathcal{E}$, but $\mathbf{X}$ remains private and inaccessible to the server.[1] Consequently, we face the challenge of performing graph learning on $\mathcal{G}$ while protecting the privacy of node features. Consistent with previous work (Sajadmanesh & Gatica-Perez, 2021; Lin et al., 2022), this paper focuses on the node classification task in graph learning. Specifically, the node set $\mathcal{V} = \mathcal{V}_l \cup \mathcal{V}_u$ is the union of the set of labeled nodes $\mathcal{V}_l$ and unlabeled ones $\mathcal{V}_u$. The label $\mathbf{y}_v$ for each node $v$ is derived from a set of possible labels, denoted $\mathcal{Y} = \{y_1, y_2, \ldots, y_c\}$. The objective of node classification (Kipf & Welling, 2017) is to learn a function $f : \mathcal{V} \to \mathcal{Y}$ that assigns labels to unlabeled nodes based on the graph structure and available node features.

### 2.2. Local Differential Privacy

LDP (Kasiviswanathan et al., 2011; Yang et al., 2024; He et al., 2025) has been extensively studied and widely deployed in decentralized data collection and analysis scenarios. In particular, major companies such as Apple (Thakurta et al., 2017), Google (Erlingsson et al., 2014), and Microsoft (Ding et al., 2017) have adopted LDP. In an LDP

---

[1] This paper concentrates on protecting node features and can be effortlessly combined with algorithms aimed at protecting neighbor lists (Zhu et al., 2023; Lin et al., 2022; Hidano & Murakami, 2024).

setting, there exists a server and multiple users, each possessing sensitive data. Users are not required to transmit their private data to an untrustworthy server. Instead, each user initially perturbs their data using a perturbation mechanism $\mathcal{M}$ and then transmits the perturbed data to the server. Following the collection of perturbed data from each user, the server performs data analysis and learning based on these data, ensuring that user privacy remains uncompromised. The formal definition of $\epsilon$-LDP is provided below.

**Definition 1** ($\epsilon$-LDP). A local perturbation mechanism $\mathcal{M}$ satisfies $\epsilon$-local differential privacy ($\epsilon$-LDP), where $\epsilon > 0$, if and only if for any user's private data $x$ and $x'$, we have:

$$\forall y \in \text{Range}(\mathcal{M}): \Pr[\mathcal{M}(x) = y] \le e^{\epsilon} \cdot \Pr[\mathcal{M}(x') = y], \quad (1)$$

where $\text{Range}(\mathcal{M})$ denotes the set of all possible outputs of the perturbation mechanism $\mathcal{M}$. In essence, LDP guarantees that the data aggregator on the server side can't reconstruct the data source, regardless of any prior knowledge. The parameter $\epsilon$, called the *privacy budget*, plays a pivotal role in balancing privacy and utility. A smaller (resp. larger) $\epsilon$ provides stronger (resp. weaker) privacy preservation but also results in lower (resp. higher) utility. LDP has several important properties, such as immunity to post-processing and sequential composition (Dwork, 2008).

### 2.3. Graph Nerual Networks

In recent years, GNNs have gained popularity for graph mining and learning. The primary goal of GNNs is to learn embeddings for each node in a graph by combining initial node features with the graph's topology. These learned node embeddings can then be applied to various downstream tasks such as node classification (Kipf & Welling, 2017; Sun et al., 2024a;b). A typical $K$-layer GNN consists of $K$ graph convolutional layers. Each layer aggregates information from neighboring nodes and updates the node's embedding. Following $K$ aggregation iterations, the embedding of a node captures information from its neighbors within $K$ hops. The formal definition of the $k$-th layer is as follows:

$$\mathbf{h}_{\mathcal{N}(v)}^{k} = \text{AGGREGATE}_{k}(\{\mathbf{h}_{u}^{k-1}, \forall u \in \mathcal{N}(v)\}), \quad (2)$$

$$\mathbf{h}_{v}^{k} = \text{UPDATE}_{k}(\mathbf{h}_{\mathcal{N}(v)}^{k}), \quad (3)$$

where $\mathcal{N}(v)$ represents the set of neighbors of node $v$ (which could include $v$ itself). For any node $u \in \mathcal{N}(v)$, $\mathbf{h}_{u}^{k-1}$ denotes the embedding of node $u$ at layer $k - 1$. The aggregation functions at layer $k$, such as mean, sum, and max, are denoted as $\text{AGGREGATE}_{k}(\cdot)$ functions. $\mathbf{h}_{\mathcal{N}(v)}^{k}$ represents the output of the aggregation function on $\mathcal{N}(v)$. $\text{UPDATE}_{k}(\cdot)$ denotes a learnable non-linear function at layer $k$, such as a neural network. Initially, $\mathbf{h}_{v}^{0} = \mathbf{x}_{v}$, indicating that the initial embedding of node $v$ is its feature vector $\mathbf{x}_{v}$.

### 2.4. Threat Model

As shown in Fig. 1, different users upload graph data to a third-party untrustworthy server. On the one hand, under a semi-honest adversary setup, although the server follows our LDP protocol honestly, it may attempt to individually learn the private information of the data owner. On the other hand, an attacker (Wang & Wang, 2022; Shen et al., 2022; Zhang et al., 2022; Meng et al., 2023) can target the GNN model to extract private information from the victim node, leading to disclosure of the user's privacy.

## 3. Methodology

In this section, we begin with a theoretical analysis of prior work on locally differentially private graph neural networks (LDPGNN) in Sec. 3.1, identifying the key factors limiting their utility. Following this, we introduce UPGNET, a utility-enhanced private graph learning model, in Sec. 3.2.

### 3.1. Theoretical Analysis of LDPGNN

A series of studies (Sajadmanesh & Gatica-Perez, 2021; Du et al., 2021; Lin et al., 2022; Jin & Chen, 2022; Qi et al., 2024; Pei et al., 2023) have been conducted on private graph learning based on LDP. However, no effort has been made to establish a unified framework for revisiting existing work to further enhance the utility of private graph learning. In this subsection, our objective is to identify the key factors influencing the aggregation estimation error by constructing and analyzing a unified node feature LDP pipeline.

#### 3.1.1. NODE FEATURE LDP PIPELINE

Currently, two state-of-the-art LDP mechanisms are applied to node features: the piecewise mechanism (PM) (Wang et al., 2019a; Pei et al., 2023) and the multi-bit mechanism (MBM) (Du et al., 2021; Sajadmanesh & Gatica-Perez, 2021; Lin et al., 2022; Jin & Chen, 2022) (for more details on PM and MBM see App. A). We propose a unified node-feature LDP pipeline that generalizes these two approaches. Assuming that the node-feature LDP mechanism is denoted as $\mathcal{M}$ and the total privacy budget employed is $\epsilon$, the node-feature LDP mechanism can be outlined in three steps:

**Perturbation.** In total, there are $|\mathcal{V}|$ users, and each user $v$ possesses a $d$-dimensional node feature $\mathbf{x}_{v}$ containing their sensitive information. Users employ an LDP mechanism $\mathcal{M}$ to protect their privacy. Initially, the mechanism $\mathcal{M}$ randomly selects $m$ dimensions from $d$ dimensions without replacement, with $m$ being a configurable parameter controlling the number of perturbed dimensions. Subsequently, each sampled dimension undergoes random perturbation with a privacy budget denoted as $\epsilon/m$, while the remaining $d - m$ dimensions are set to 0. Ultimately, $\mathbf{x}_{v}$ undergoes the $\mathcal{M}$ mechanism to yield $\mathbf{x}_{v}'$, denoted as $\mathbf{x}_{v}' = \mathcal{M}(\mathbf{x}_{v})$.

**Calibration.** Following perturbation, $\mathbf{x}'_v$ remains biased, i.e., $\mathbb{E}\left[\mathbf{x}'_v\right] \neq \mathbf{x}_v$. Let $\boldsymbol{\sigma} = \mathbb{E}\left[\mathbf{x}'_v\right] - \mathbf{x}_v$, representing the expected bias shift. The server calibrates the perturbed values with $\boldsymbol{\sigma}$. Note that $\boldsymbol{\sigma} = 0$ indicates an unbiased estimate.

**Aggregation.** After receiving the feature vectors $\mathbf{x}'_v$ for all users $v \in \mathcal{V}$, the server aggregates these vectors as follows:

$$\widehat{\mathbf{h}}_{\mathcal{N}(v)} = \text{AGGREGATE}\left(\{\mathbf{x}'_u, \forall u \in \mathcal{N}(v)\}\right), \quad (4)$$

where $\widehat{\mathbf{h}}_{\mathcal{N}(v)}$ represents the estimated embedding for any given node $v$ after undergoing the $\text{AGGREGATE}(\cdot)$. This estimate is derived by aggregating the perturbed node feature vectors $\mathbf{x}'_u$ from all nodes $u$ adjacent to the target node $v$.

**Theorem 2.** *Assuming $\boldsymbol{\sigma} = 0$, the aggregator function defined by Eq. (4) is an unbiased estimate, i.e., for any $v$,*

$$\mathbb{E}[\widehat{\mathbf{h}}_{\mathcal{N}(v)}] = \mathbf{h}_{\mathcal{N}(v)}. \quad (5)$$

As demonstrated in Thm. 2, the aggregation process is an unbiased estimate when $\boldsymbol{\sigma} = 0$ and the AGGREGATE function is linear, meaning the output is a weighted summation of the inputs. For the proof, please refer to App. B.1.

### 3.1.2. KEY FACTOR ANALYSIS

In Sec. 3.1.1, we establish an LDP analytical pipeline for the node features, consisting of three stages: perturbation, calibration, and aggregation. In the context of privacy-preserving graph learning, the aggregation stage significantly impacts the overall utility of graph learning. High-quality aggregation helps mitigate the injected noise to a greater extent. Therefore, our objective is to explore the key factors that directly influence the estimation error in the aggregation stage. The estimation error is defined as the discrepancy between $\widehat{\mathbf{h}}_{\mathcal{N}(v)}$, obtained by aggregating the perturbed node features $\mathbf{x}'$, and $\mathbf{h}_{\mathcal{N}(v)}$, obtained by aggregating the original node features $\mathbf{x}$. This discrepancy is represented as $\xi_i = |(\widehat{\mathbf{h}}_{\mathcal{N}(v)})_i - (\mathbf{h}_{\mathcal{N}(v)})_i|, i \in \{1, \ldots, d\}$. Based on Bernstein's inequality (Giroux et al., 1979), Thm. 3 provides an analysis of the various factors influencing the discrepancy.

**Theorem 3.** *Given the aggregator for the first layer and $\delta > 0$, with probability at least $1 - \delta$, for any node $v$, we have:*

$$\max \xi_i = \mathcal{O}(\sqrt{d \log(d/\delta)}/(\epsilon\sqrt{|\mathcal{N}(v)|})), i \in \{1, \ldots, d\}. \quad (6)$$

According to Thm. 3, after eliminating the known privacy budget parameter $\epsilon$ and the analysis parameter $\delta$, two key factors that impact the estimation error are *the feature dimension $d$* and *the neighborhood size $|\mathcal{N}(v)|$*. From Eq. (6), we infer that a smaller effective $d$ is more conducive to reducing the estimation error, while a larger effective $|\mathcal{N}(v)|$ is also advantageous to minimizing the estimation error. Therefore, in Sec. 3.2, our objective is enhance the utility of privacy-preserving graph learning by influencing $d$ and $|\mathcal{N}(v)|$. Please see App. B.2 for the proof of Thm. 3.

## 3.2. UPGNET: Utility-Enhanced Private GNNs

In this section, we propose a utility-enhanced private graph learning framework called UPGNET. Designed for various node feature LDP protocols, UPGNET incorporates plug-and-play NFR and HOA layers that significantly boost utility. Fig. 2 (a) provides an overview of UPGNET, and Fig. 2 (b) illustrates the two architectures of UPGNET.

### 3.2.1. OVERVIEW

This subsection introduces the foundational design principles of UPGNET. According to Thm. 3, the key factors influencing the estimation error during the aggregation process are the feature dimension and the neighborhood size. Naturally, our objective is to minimize the estimation error during aggregation by targeting these two crucial factors, thereby enhancing the practicality of private graph learning. To this end, we design from the following two perspectives:

**Expanding the Effective Neighborhood Size.** In order to extend the effective neighborhood size, direct multi-layer aggregation is a potential approach. However, our analysis reveals that this method is significantly limited by over-smoothing (Rusch et al., 2023), which adversely affected the denoising performance. To address this, we propose a *Higher-Order Aggregator (HOA) layer*, leveraging personalized aggregation and Dirichlet energy analysis to effectively mitigate over-smoothing and reduce noise injection.

**Reducing the Effective Feature Dimensions.** In order to minimize estimation error by reducing the effective feature dimensions, we focus on the aggregation stage[2]. Based on $L_1$-regularization and proximal gradient descent (PGD) (Nitanda, 2014; Li & Lin, 2015; Duan et al., 2022), we introduce the *Node Feature Regularizer (NFR) layer*.

Through theoretical analysis and experimental validation, UPGNET, in both the H-N (see Sec. 3.2.2) and the N-H architecture (see Sec. 3.2.3), effectively integrates the HOA layer and the NFR layer to reduce estimation error and enhance learning utility across various LDP mechanisms.

### 3.2.2. UPGNET IN H-N ARCHITECTURE

In this part, we introduce the H-N architecture of UPGNET, where the perturbed node features $\mathbf{x}'$ are successively enhanced by the HOA layer followed by the NFR layer.

**Higher-Order Aggregator.** According to Thm. 3, as the size of $\mathcal{N}(v)$ (i.e., $|\mathcal{N}(v)|$) increases, the estimation error decreases at a rate proportional to the square root of the node degree. This indicates that larger $|\mathcal{N}(v)|$ results in

---

[2]The reason for targeting the aggregation stage, rather than the perturbation or calibration stages, is that designing from the aggregation stage provides a certain level of generality and independence from different LDP mechanisms.

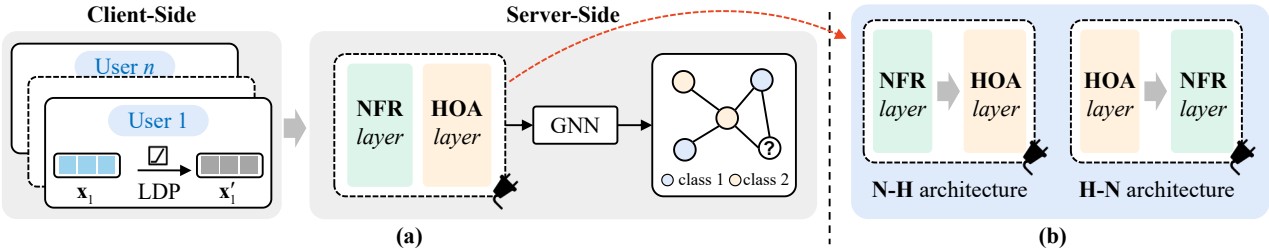

*Figure 2.* (a) Overview of our proposed UPGNET. On the client side, local node features $\mathbf{x}$ are perturbed using LDP to obtain $\mathbf{x}'$. On the server side, $\mathbf{x}'$ undergoes processing through an *Node Feature Regularization (NFR) layer* and an *High-Order Aggregator (HOA) layer* to enhance utility before being input into the GNN for graph learning, enabling downstream tasks such as node classification. (b) UPGNET features two distinct architectures: H-N (*HOA layer* followed by *NFR layer*) and N-H (*NFR layer* followed by *HOA layer*).

smaller estimation errors. However, in practical scenarios, $|\mathcal{N}(v)|$ is usually quite small. One approach to address this issue is to expand $\mathcal{N}(v)$ by directly aggregating multiple layers of node features across $K$ hops (Abu-El-Haija et al., 2019; Gasteiger et al., 2019; Chen et al., 2020; Sajadmanesh & Gatica-Perez, 2021; Lin et al., 2022) (defined as the SKA method). However, the SKA scheme encounters the over-smoothing (Rusch et al., 2023), where increasing the value of $K$ causes node embeddings to converge, reducing the effectiveness of information aggregation from higher-order neighbors and skewing the calibration of aggregation errors. To better understand the over-smoothing issue, we examine the Dirichlet energy (Rusch et al., 2023) of the estimated node embeddings on the graph, a primary measure of over-smoothing in deep GNNs. Specifically, the estimated embedding $\widehat{\mathbf{h}}$ is modeled as a combination of the original embedding $\mathbf{h}$ and the noise signal $\eta$, i.e., $\widehat{\mathbf{h}} = \mathbf{h} + \eta$. The Dirichlet energy $\Upsilon(\cdot)$ of $\widehat{\mathbf{h}}$ is then defined as:

$$\Upsilon(\widehat{\mathbf{h}}) = \frac{1}{|\mathcal{V}|} \sum_{i \in \mathcal{V}} \sum_{j \in \mathcal{N}(v_i)} \left( \left\| \mathbf{h}_i^k - \mathbf{h}_j^k \right\|_2^2 + \underbrace{\left\| \eta_i^k - \eta_j^k \right\|_2^2}_{\text{noise signal}} \right), \quad (7)$$

where $k \in \{1, 2, \cdots, K\}$ represents the step parameter. Eq. (7) highlights two key observations: ① *Over-smoothing exists.* As $k$ increases, the difference between the embedding $\mathbf{h}_i$ and its neighbor embedding $\mathbf{h}_j$ diminishes. After several rounds of propagation, all node features converge, causing a sharp decline in the first term of $\Upsilon(\widehat{\mathbf{h}})$, resulting in the typical oversmoothing phenomenon. ② *Noise exacerbates over-smoothing.* Under the LDP, when $\epsilon$ approaches 0, the noise term $\|\eta_i - \eta_j\|_2^2$ dominates the Dirichlet energy. This accelerates the homogenization of node features, leading to faster energy decay and thus intensifying the oversmoothing effect. Therefore, mitigating the over-smoothing problem to extend more effective neighborhood sizes is crucial.

To address the aforementioned issues, we propose a utility-enhanced graph convolution layer called the *High-Order Aggregator (HOA)*, as shown in Alg. 1. Compared to SKA, it has two advantages in reducing estimation error for noisy data: ① mitigating over-smoothing and ② reducing noise

---

**Algorithm 1** High-Order Aggregator (HOA) Layer

**Input:** $\mathcal{G} = (\mathcal{V}, \mathcal{E})$, input vector $\mathbf{x}_v, \forall v \in \mathcal{V}$, step parameter $K \geq 0$, linear aggregator function AGGREGATE($\cdot$)
**for** $v \in \mathcal{V}$ **do**
  $\mathbf{h}_{\mathcal{N}(v)}^0 = 0, \mathbf{x}_{\mathcal{N}(v)}^0 = \mathbf{x}_v$
  **for** $k = 1$ **to** $K$ **do**
    $\mathbf{x}_{\mathcal{N}(v)}^k = \text{AGGREGATE}\left( \{ \mathbf{x}_{\mathcal{N}(u)}^{k-1}, \forall u \in \mathcal{N}(v) \backslash \{v\} \} \right)$
    $\mathbf{h}_{\mathcal{N}(v)}^k = \mathbf{h}_{\mathcal{N}(u)}^{k-1} + \mathbf{x}_{\mathcal{N}(v)}^k$
  **end for**
  $\mathbf{h}_v = \frac{1}{K} \mathbf{h}_{\mathcal{N}(v)}^K$
**end for**
**Return:** Aggregated embedding vector $\mathbf{h}_v, \forall v \in \mathcal{V}$

---

bias injection. First, as shown in Thm. 4, the energy ratio $\Phi_K$ between HOA($\cdot$) and SKA($\cdot$) approaches 0 as $K \to \infty$. This indicates that HOA is less influenced by information from infinite-hop receptive fields, allowing it to mitigate over-smoothing and expand into larger neighborhoods while still aggregating data from smaller ones. Second, HOA employs personalized weightings during neighbor information aggregation, assigning the highest weight to the nearest neighbor. The insight behind our approach is that information from the closest neighbor is more effective in calibrating noise for that specific node, justifying its higher weight, while more distant neighbors receive lower weights. This design further alleviates noise bias injection. See App. B.3 for the proof of Thm. 4 and more details.

**Theorem 4.** *Let* $\Upsilon_{HOA}^k$ *and* $\Upsilon_{SKA}^k$ *represent the Dirichlet energies of* HOA($\cdot$) *and* SKA($\cdot$) *at layer* $k$, *respectively. The energy ratio of* HOA($\cdot$) *to* SKA($\cdot$) *across* $K$ *layers satisfies:*

$$\Phi_K = \lim_{K \to \infty} \left( \sum_{k=1}^K \Upsilon_{HOA}^k \bigg/ \sum_{k=1}^K \Upsilon_{SKA}^k \right) = 0. \quad (8)$$

As in (Sajadmanesh & Gatica-Perez, 2021), we also employ the GCN aggregator function (Kipf & Welling, 2017) and perform the aggregations without including the self-loop, which will facilitate the reduction of the total noise.

**Node Feature Regularization**. Regularization is a clas-

sical method for optimizing minimization tasks (Hoyer, 2004; Bühlmann & Van De Geer, 2011; Negahban et al., 2017; Duan et al., 2022). Among various regularization techniques, $L_1$-regularization tends to produce sparse solutions. In other words, parameters obtained through $L_1$-regularization are more likely to have fewer non-zero components. This property facilitates the implementation of embedded feature selection, aligning with our objective of reducing the effective feature dimension. Next, we formalize the $L_1$-regularization problem within UPGNET under the H-N architecture. We employ the mean aggregation function and define the embedding of node $v$ after the server aggregates the perturbed feature vectors as follows:

$$\widehat{\mathbf{h}}_v = \text{AGGREGATE} \left( \{ \mathbf{x}'_u, \forall u \in \mathcal{N}(v) \} \right) = \frac{1}{|\mathcal{N}(v)|} \sum_{u \in \mathcal{N}(v)} \mathbf{x}'_u, \quad (9)$$

where $\mathbf{x}'_u$ represents the noisy node features of node $u$. We define the loss function $\mathcal{L}_1(\boldsymbol{w})$ as follows:

$$\mathcal{L}_1(\boldsymbol{w}) = \frac{1}{2 |\mathcal{N}(v)|} \cdot \sum_{u \in \mathcal{N}(v)} \| \mathbf{x}'_u - \boldsymbol{w} \|_2^2. \quad (10)$$

Based on this, we add the $L_1$-regularization terms $\| \boldsymbol{w} \|_1$ to $\mathcal{L}_1(\boldsymbol{w})$ to obtain the enhanced node embedding $\widetilde{\mathbf{h}}_v$ for any node $v$ as $\widetilde{\mathbf{h}}_v = \arg\min_{\boldsymbol{w} \in \mathbb{R}^d} \mathcal{L}_1(\boldsymbol{w}) + \mu_1 \| \boldsymbol{w} \|_1$. Thm. 5 derives the above $L_1$-regularization problem by employing the *proximal gradient descent* (PGD) (Nitanda, 2014; Li & Lin, 2015; Duan et al., 2022) method.

**Theorem 5.** *For any node $v \in \mathcal{V}$ and any feature dimension $i \in \{1, \ldots, d\}$, $(\widetilde{\mathbf{h}}_v)_i$ in the following equation can efficiently achieve feature selection for $(\widehat{\mathbf{h}}_v)_i$:*

$$(\widetilde{\mathbf{h}}_v)_i = \text{sign} \left( (\widehat{\mathbf{h}}_v)_i \right) \cdot \max \left( |(\widehat{\mathbf{h}}_v)_i| - \mu_1, 0 \right), \quad (11)$$

where $\text{sign}(\cdot)$ denotes the sign function, which takes 1 if $(\widehat{\mathbf{h}}_v)_i > 0$, 0 if $(\widehat{\mathbf{h}}_v)_i = 0$, and -1 if $(\widehat{\mathbf{h}}_v)_i < 0$. The optimal value for $\mu_1$ is $\tau_1 B / \bar{d}^K$, where $\tau_1 \in (0, 1)$, with $B$ as the boundary of the perturbed node features, $\bar{d}$ as the approximate average degree of the graph, and $K$ as the step parameter of the HOA layer. By Thm. 5, we conclude that Eq. (11) can efficiently achieve feature selection for $\widehat{\mathbf{h}}_v$, thereby contributing to enhancing the utility of private graph learning. See App. B.4 for the proof.

### 3.2.3. UPGNET IN N-H ARCHITECTURE

Under the N-H architecture of UPGNET, the HOA is consistent with Alg. 1. The NFR specifically aims to enhance utility through efficient feature selection of the perturbed node features $\mathbf{x}'$ directly using $L_1$-regularization. The objective function $\mathcal{L}_2$ of HOA is formalized as follows:

$$\mathcal{L}_2(\mathbf{x}) = \frac{1}{2} \| \mathbf{x}' - \mathbf{x} \|_2^2 + \mu_2 \| \mathbf{x} \|_1. \quad (12)$$

*Table 1.* Statistics of graph datasets.

| Dataset | Classes | Nodes | Edges | Features | Deg. |
|---|---|---|---|---|---|
| Cora | 7 | 2,708 | 5,278 | 1,433 | 3.90 |
| Citeseer | 6 | 3,327 | 4,552 | 3,703 | 2.74 |
| LastFM | 18 | 7,624 | 27,806 | 7,842 | 7.29 |
| Facebook | 4 | 22,470 | 170,912 | 4,714 | 15.21 |

*Deg.* in the table denotes the average node degree.

**Theorem 6.** *For any node $v$ and any $i \in \{1, \ldots, d\}$, $(\widetilde{\mathbf{x}}_v)_i$ in Eq. (13) can efficiently achieve feature selection:*

$$(\widetilde{\mathbf{x}}_v)_i = \text{sign} \left( (\mathbf{x}'_v)_i \right) \cdot \max \left( |(\mathbf{x}'_v)_i| - \mu_2, 0 \right), \quad (13)$$

where the optimal value for $\mu_1$ is $\tau_2 B$, where $\tau_2 \in (0, 1)$, with $B$ as the boundary of the perturbed node features. According to Thm. 6, we conclude that Eq. (13) can efficiently achieve feature selection for $\mathbf{x}'_v$, thus enhancing the utility of graph learning. See Sec. C and App. B.5 for more details.

### 3.2.4. PRIVACY AND COMPLEXITY ANALYSIS

**Privacy Analysis.** The PM and MBM satisfy $\epsilon$-LDP for each node. The entire training process remains LDP compliant due to the robustness of DP against the post-processing theorem (Dwork et al., 2014). Moreover, any subsequent prediction is bounded by the post-processing theorem (Dwork et al., 2014), since the LDP protocol is applied only once to the private data. This ensures that LDP holds for all nodes throughout the process. For more details, see App. D.

**Complexity Analysis.** The computational complexity of UPGNET mainly arises from its two key components: the HOA and the NFR. Through analysis, the overall complexity of UPGNET is $\mathcal{O}(K \cdot |\mathcal{E}| \cdot d + |\mathcal{V}| \cdot d)$, scaling linearly with graph size and feature dimensionality. This ensures that UPGNET remains both practical and scalable for large-scale graphs with high-dimensional data. See App. E for more detailed analysis and comparisons with baselines.

## 4. Experiments

In this section, we conduct a series of experiments to validate the performance of UPGNET and its core components. More experimental results can be found in App. F.

### 4.1. Experimental Setting

**Datasets.** We conduct experiments on four representative graph datasets: Cora (Yang et al., 2016), Citeseer (Yang et al., 2016), LastFM (Rozemberczki & Sarkar, 2020), and Facebook (Rozemberczki et al., 2021). These datasets are commonly used in graph machine learning (Wu et al., 2020; Zhang et al., 2020). Table 1 provides the statistics for these datasets, with specific descriptions as follows:

- *Cora* and *CiteSeer*. They are well-known citation networks, where each node represents a scientific paper, and

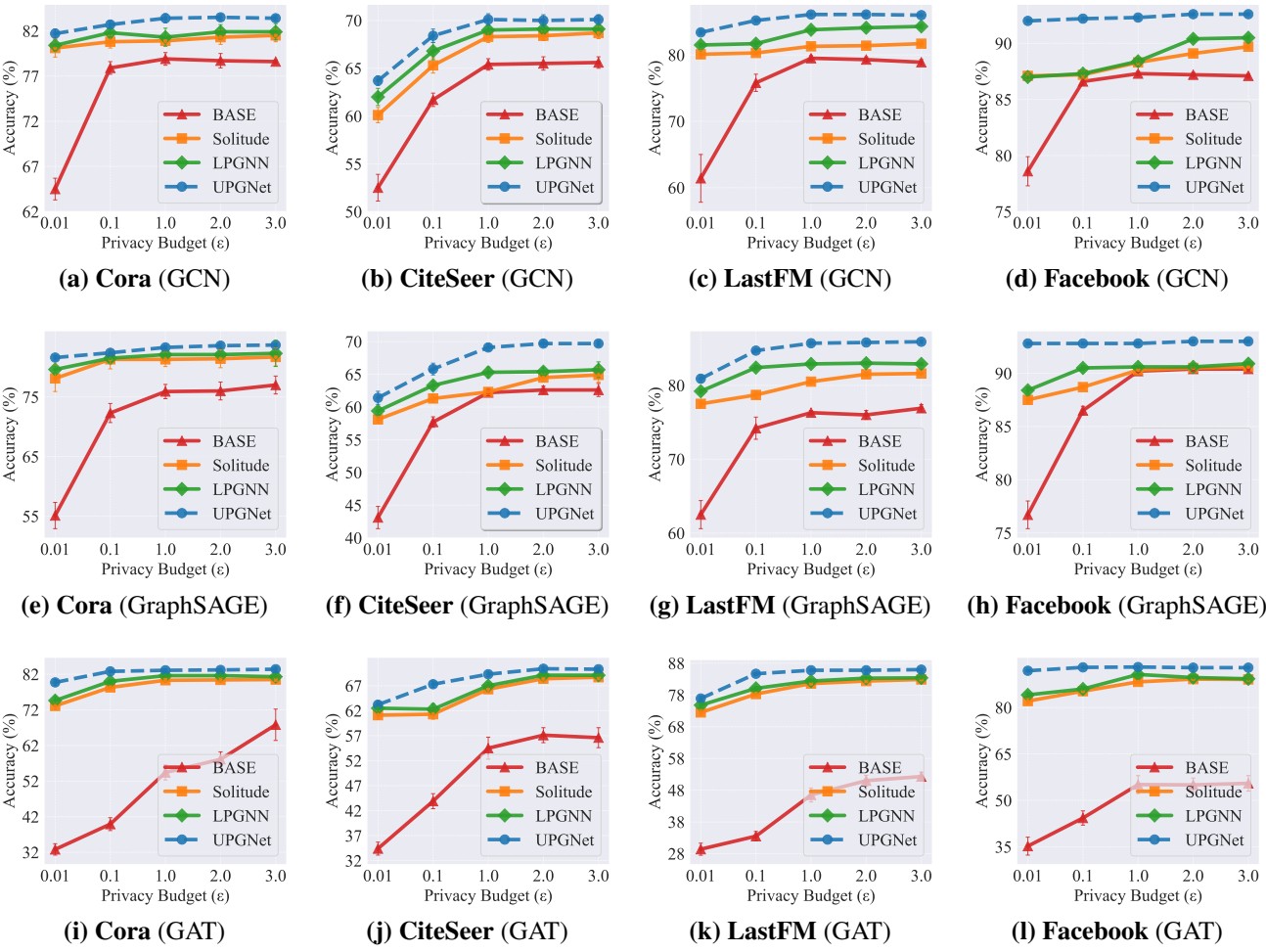

*Figure 3.* Performance of UPGNET and other baselines. X-axis represents $\epsilon$ and y-axis represents test accuracy (%).

the edges denote citation links. Each node contains a bag-of-words feature vector and a label for each category.

- *Facebook*. This social network consists of nodes as official Facebook pages, with edges representing mutual liking relationships. Each node has a feature extracted from the site description and a label indicating the category.

- *LastFM*. Nodes in this dataset represent users of the music streaming service LastFM and links represent friendships between them. The classification task is to predict the users' home country given the artists liked them.

**Baselines.** To comprehensively assess the performance of UPGNET, we compare it with the following baselines: The *NonPriv* sets $\epsilon = \infty$ and inputs clean (non-perturbed) node features directly into the GNN for graph learning. In contrast, *BASE* utilizes the GNN for graph learning directly after using node feature LDP protocols to perturb feature vectors, without incorporating additional utility enhancement strategies. *LPGNN* (Sajadmanesh & Gatica-Perez,

2021) and *Solitude* (Lin et al., 2022) apply different strategies to achieve locally differentially private graph learning. In addition, we consider the multi-bit mechanism (MBM) (Sajadmanesh & Gatica-Perez, 2021) and the piecewise mechanism (PM) (Wang et al., 2019a) independently.

**Parameter Settings.** All datasets are randomly divided into 50/25/25% for training, validation, and test sets, respectively. To evaluate the performance of UPGNET, we use three representative GNN architectures, graph convolutional networks (GCN) (Kipf & Welling, 2017), GraphSAGE (Hamilton et al., 2017), and graph attention networks (GAT) (Velickovic et al., 2018) as backbone models. By default, the dataset used is Cora, the LDP protocol applied is the MBM, the GNN model is the GCN, and UPGNET adopts the N-H architecture. For more details, see App. F.1 and F.2.

**Evaluation Metrics.** Consistent with prior work (Sajadmanesh & Gatica-Perez, 2021; Lin et al., 2022), we conduct experiments on the node classification task, using classification accuracy as the primary metric to evaluate the per-

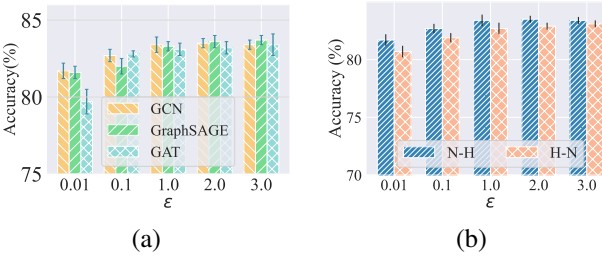

(a)                      (b)

*Figure 4.* (a) Comparison of the performance of UPGNET under different GNN models. (b) Comparison of the performance of UPGNET in H-N vs. N-H architectures.

formance of UPGNET. All models undergo 500 training iterations and the best model is chosen for testing based on validation loss. Accuracy is measured over 10 consecutive runs, and we report the average along with 95% confidence intervals calculated by bootstrapping over 1000 samples.

### 4.2. Evaluating the Performance of UPGNET

In this experiment, we vary $\epsilon$ in $\{0.01, 0.1, 1.0, 2.0, 3.0\}$ to thoroughly validate the performance under different noise scales. The experimental results across four datasets and various GNN backbone models are presented in Fig. 3. It shows that, in all cases, UPGNET consistently achieves higher accuracy than BASE, LPGNN and Solitude, and in some instances, it even approaches the accuracy of NonPriv. For example, in the case of Fig. 3 (f), UPGNET increases accuracy by about 5% over LPGNN and Solitude when $\epsilon = 3.0$. Moreover, for Facebook (Fig. 3 (d), (h)), the accuracy of UPGNET is closely aligned with that of NonPriv. All these observations underscore the superior utility of UPGNET.

### 4.3. Comparison of Different GNN Models

Fig. 4(a) intuitively compares the accuracy of UPGNET under different privacy budgets $\epsilon$ and across various GNN models. The results reveal that the GAT slightly worse than GCN and GraphSAGE. Specifically, under high privacy settings (*e.g.*, when $\epsilon = 0.01$), the accuracy gap between GAT and the other two models becomes more pronounced. The GAT model introduces an attention mechanism that learns attention coefficients to weight the neighbors in neighborhood aggregation. Consequently, this makes GAT more sensitive to feature perturbations, resulting in a greater degradation of utility in high-noise settings. On the other hand, the utility of GAT remains comparable when $\epsilon \geq 0.1$.

### 4.4. Ablation Study on the Performance of NFR

In this experiment, we investigate the utility enhancement of Node Feature Regularizer (NFR) layer for two node feature LDP mechanisms: piecewise mechanism (PM) and multi-bit mechanism (MBM). Table 2 shows the accuracy comparison

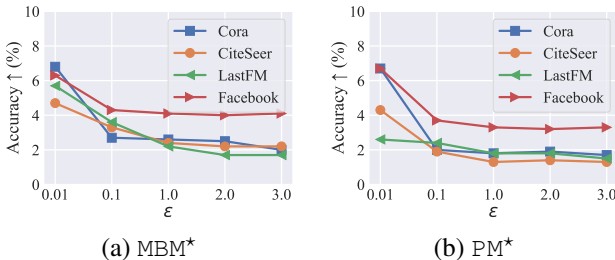

(a) MBM$^\star$               (b) PM$^\star$

*Figure 5.* Comparison of the utility enhancement of MBM$^\star$ and PM$^\star$ under different $\epsilon \in \{0.01, 0.1, 1.0, 2.0, 3.0\}$. X-axis represents privacy budget $\epsilon$ and y-axis represents test accuracy enhancement.

*Table 2.* Applying NFR layer to node feature LDP mechanisms PM and MBM to achieve utility enhancement. MBM$^\star$ and PM$^\star$ represent the integration of the NFR layer with MBM and PM, respectively.

| Dataset | Mech. | $\epsilon = 0.01$ | $\epsilon = 0.1$ | $\epsilon = 1.0$ | Overall |
|---------|-------|-------------------|------------------|------------------|---------|
| Cora | MBM | 64.5 | 77.9 | 78.9 | - |
| | MBM$^\star$ | 71.3 ↑6.8 | 80.6 ↑2.7 | 81.5 ↑2.6 | ↑**4.0** |
| | PM | 65.3 | 78.5 | 79.4 | - |
| | PM$^\star$ | 72.0 ↑6.7 | 80.5 ↑2.0 | 81.2 ↑1.8 | ↑**3.5** |
| CiteSeer | MBM | 52.5 | 61.7 | 65.4 | - |
| | MBM$^\star$ | 57.2 ↑4.7 | 65.0 ↑3.3 | 67.8 ↑2.4 | ↑**3.5** |
| | PM | 53.1 | 62.8 | 66.1 | - |
| | PM$^\star$ | 57.4 ↑4.3 | 64.7 ↑1.9 | 67.4 ↑1.3 | ↑**2.5** |
| LastFM | MBM | 61.4 | 75.8 | 79.5 | - |
| | MBM$^\star$ | 67.1 ↑5.7 | 79.4 ↑3.6 | 81.7 ↑2.2 | ↑**3.8** |
| | PM | 66.1 | 77.1 | 80.1 | - |
| | PM$^\star$ | 68.7 ↑2.6 | 79.5 ↑2.4 | 81.9 ↑1.8 | ↑**2.3** |
| Facebook | MBM | 78.6 | 86.6 | 87.3 | - |
| | MBM$^\star$ | 84.9 ↑6.3 | 90.9 ↑4.3 | 91.4 ↑4.1 | ↑**4.9** |
| | PM | 78.4 | 88.4 | 88.5 | - |
| | PM$^\star$ | 85.1 ↑6.7 | 90.8 ↑3.7 | 91.8 ↑3.3 | ↑**4.6** |

↑ indicates the increase in utility after incorporating the NFR.

between MBM and MBM$^\star$ (MBM + NFR layer) as well as PM and PM$^\star$ (PM + NFR layer) across different datasets and $\epsilon$. The results in Table 2 clearly demonstrate that applying the NFR layer improves graph learning accuracy in all cases. Moreover, Table 2 and Fig. 5 indicates that NFR layer is more effective in improving graph learning accuracy when the privacy budget is small compared to when the privacy budget is large. For instance, in the case of the Cora dataset with MBM perturbing the node features, when $\epsilon = 0.01$, MBM$^\star$ improves accuracy by approximately 7% over MBM. When $\epsilon = 1.0$, MBM$^\star$ enhances accuracy by 2.6% over MBM. This is attributed to the fact that when $\epsilon$ is small, more noise is injected into the node features, so our NFR layer calibrates the noise and improves the accuracy more significantly.

### 4.5. Ablation Study on the Performance of HOA

In this experiment, we investigate two aspects: first, whether Higher-Order Aggregator (HOA) layer can mitigate the over-smoothing in multilayer aggregation; and second, whether the HOA can effectively enhance the performance of private graph learning. To achieve these objectives, we compare

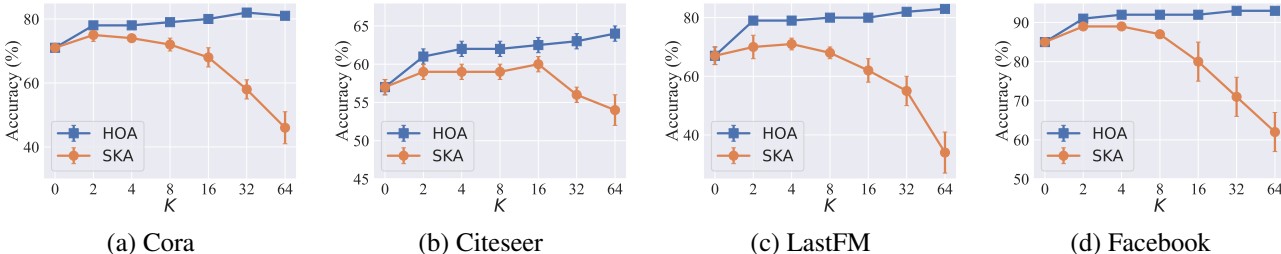

*Figure 6.* Effect of HOA vs. SKA on graph learning performance across various steps $K \in \{0, 2, 4, 8, 16, 32, 64\}$.

HOA with SKA and set $K = \{0, 2, 4, 8, 16, 32, 64\}$ for both HOA and SKA, consider the privacy budget $\epsilon = 0.01$. Fig. 6 shows the performance of HOA compared to SKA on accuracy for different values of $K$. As illustrated in Fig. 6, for all datasets, the accuracy trend under SKA initially rises with increasing $K$, but after a certain point, it sharply declines, while the accuracy under HOA continues to improve steadily. For example, in the LastFM dataset (Fig. 6 (c)), SKA's accuracy peaks at $K = 4$ but then drops rapidly, falling below 40% accuracy at $K = 64$. This sharp decline is due to the oversmoothing effect in SKA, where larger $K$ values cause node embeddings to become overly similar, diminishing graph learning performance. In contrast, the HOA algorithm shows a steady increase in accuracy as $K$ grows, indicating that the proposed HOA layer successfully mitigates oversmoothing and effectively aggregates useful information from expanded neighborhoods. Furthermore, for $K \in \{2, 4, 8, 16, 32, 64\}$, HOA consistently outperforms SKA, demonstrating that the HOA can significantly enhance the learning utility. See App. F.3 and F.5 for more details.

### 4.6. Comparison of Different Architectures

We evaluate the effect of both the N-H and H-N architectures on utility by varying $\epsilon \in \{0.01, 0.1, 1.0, 2.0, 3.0\}$ and conducting comparisons using the GCN. As shown in Fig. 4(b), for the Cora, the N-H architecture outperforms the H-N slightly in terms of accuracy. Notably, with smaller $\epsilon$, the N-H architecture excels in feature dimension optimization, which allows it to better handle the information loss from noise injection. However, as $\epsilon$ increases, the performance gap between the N-H and H-N architectures narrows, indicating that the early application of the NFR layer in the N-H architecture is more effective in expanding the neighborhood range and enhancing utility when the noise is higher.

## 5. Related work

Recently, a series of works related to locally differentially private GNNs have been proposed. Sajadmanesh & Gatica-Perez (2021) propose a privacy-preserving graph learning framework called LPGNN, which assumes that node features are private and the server has access to the graph topol-

ogy, aligning with the scenario of this paper. In LPGNN, each user perturbs their features using the multi-bit mechanism (MBM). However, this paper demonstrates that MBM introduces excessive noise to the feature vector, thereby reducing the utility of the final private graph learning process. Similarly, Du et al. (2021), Lin et al. (2022) and Jin & Chen (2022) utilize the MBM to perturb node features or employ the SKA (Sajadmanesh & Gatica-Perez, 2021; Lin et al., 2022) to calibrate noisy features. Besides MBM, the PM (Wang et al., 2019a; Pei et al., 2023) has also been explored for privacy-preserving graph learning. Yet, it faces similar utility challenges. In contrast to these approaches, our paper introduces a utility-enhanced locally private graph learning framework applicable to various node feature perturbation mechanisms, including MBM and PM, to further enhance the utility of private graph learning.

In addition to node feature perturbation, other efforts have addressed link privacy under LDP. Lin et al. (2022) adopted a naive randomized response mechanism (Qin et al., 2017) to protect adjacency lists. Hidano & Murakami (2024) propose a link LDP mechanism called DPRR, which injects noise into adjacency lists and node degrees respectively and calibrates the noisy links by a degree-sampling method. Zhu et al. (2023) propose a Bayesian estimation-based link LDP mechanism called BLINK. Similar to DPRR, BLINK injects noise into adjacency lists and node degrees, and then estimates the ground truth graph using the adjacency lists as prior and the node degrees as evidence. This work, in contrast, focuses on protecting node features, and is orthogonal to these approaches. It can be seamlessly integrated with existing methods designed to protect neighbor lists.

## 6. Conclusion

In this paper, we initially establish a pipeline to generalize the LDP protocols for perturbing node features. Through our analysis, we identify two key factors that affect the estimation error. Building on these insights, we propose UPGNET, which incorporates NFR and HOA layers. The generalization and effectiveness of UPGNET and its components are validated through theoretical analysis and extensive experiments in various datasets and parameter settings.

## Acknowledgements

This work is supported in part by the National Natural Science Foundation of China (62072052), in part by the National Natural Science Foundation of China (62372051).

## Impact Statement

This paper presents work aiming to advance the field of privacy-preserving graph learning. It focuses on enhancing the utility of private graph learning while preserving the privacy of graph data using local differential privacy. Our approach introduces a more efficient framework for privacy-preserving graph learning, enhancing the overall utility of learning tasks while ensuring that user data remains protected from potential privacy breaches. There are many potential societal consequences of our work, none which we feel must be specifically highlighted here. We believe this work will contribute to the development of more efficient and scalable privacy-preserving graph learning frameworks.

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

# A. Node Feature LDP Mechanisms

## A.1. Piecewise Mechanism

The one-dimensional piecewise mechanism (PM) (Wang et al., 2019a) takes $x \in [-1, 1]$ as input[3] and outputs the perturbed value $x' \in [-Q, Q]$, where $Q = \frac{e^{\epsilon/2}+1}{e^{\epsilon/2}-1}$. The perturbed value is sampled from the following distribution:

$$\Pr[x' = c | x] = \begin{cases} p, & \text{if } c \in [l(x), r(x)] \\ \frac{p}{e^\epsilon}, & \text{if } c \in [-Q, l(x)) \cup (r(x), Q] \end{cases}, \tag{14}$$

where $p = \frac{e^\epsilon - e^{\epsilon/2}}{2e^{\epsilon/2}+2}$, $l(x) = \frac{Q+1}{2} \cdot x - \frac{Q-1}{2}$, and $r(x) = l(x) + Q - 1$. In its high-dimensional form, the piecewise mechanism initially uniformly samples $m$ dimensions from the $d$ dimensions without replacement. Subsequently, it perturbs the data in each of the selected dimensions using the $\epsilon/m$-LDP. According to the sequential composition property (Dwork, 2008) of the LDP mechanism, it adheres to $\epsilon$-LDP. After receiving all the perturbed data, the server performs de-biasing on the reporting data $x^*$ using the following equation: $x' = \frac{d}{m} \cdot x^*$. The variance of $x'$ is as follows:

$$Var[x'] = \frac{d(e^{\epsilon/2m}+3)}{3m(e^{\epsilon/2m}-1)^2} + \left[ \frac{d \cdot e^{\epsilon/2m}}{m(e^{\epsilon/2m}-1)} - 1 \right] \cdot x^2. \tag{15}$$

The range of $x'$ is as follows:

$$x' \in \left[ -\frac{d}{m} \cdot \frac{e^{\epsilon/2}+1}{e^{\epsilon/2}-1}, \frac{d}{m} \cdot \frac{e^{\epsilon/2}+1}{e^{\epsilon/2}-1} \right]. \tag{16}$$

## A.2. Multi-bit Mechanism

The multi-bit mechanism (MBM) (Sajadmanesh & Gatica-Perez, 2021) is an extension of the 1-bit mechanism (Ding et al., 2017). In its one-dimensional form, for any original data $x \in [-1, 1]$, the distribution followed by the perturbed value $x' \in \{-1, 1\}$ is as follows:

$$\Pr[x' = c | x] = \begin{cases} \frac{1}{e^\epsilon+1} + \frac{x+1}{2} \cdot \frac{e^\epsilon-1}{e^\epsilon+1}, & \text{if } c = 1 \\ \frac{e^\epsilon}{e^\epsilon+1} - \frac{x+1}{2} \cdot \frac{e^\epsilon-1}{e^\epsilon+1}, & \text{if } c = -1 \end{cases}. \tag{17}$$

In its high-dimensional form, similar to the piecewise mechanism, each sampled dimension carries out $\epsilon/m$-LDP perturbation. After receiving all the perturbed data, the server transforms the reporting data $x^*$ to its unbiased estimate $x' = \frac{d}{m} \cdot \frac{e^{\epsilon/m}+1}{e^{\epsilon/m}-1} \cdot x^*$. The variance of $x'$ is as follows:

$$Var[x'] = \frac{d}{m} \cdot \left( \frac{e^{\epsilon/m}+1}{e^{\epsilon/m}-1} \right)^2 - x^2. \tag{18}$$

The range of $x'$ is as follows:

$$x' \in \left\{ -\frac{d}{m} \cdot \frac{e^{\epsilon/m}+1}{e^{\epsilon/m}-1}, 0, \frac{d}{m} \cdot \frac{e^{\epsilon/m}+1}{e^{\epsilon/m}-1} \right\}. \tag{19}$$

# B. Theoretical Proof

## B.1. Proof for Theorem 2

*Proof.* According to Eq. (4), we have: $\mathbb{E}\left[\widehat{\mathbf{h}}_{\mathcal{N}(v)}\right] = \mathbb{E}\left[\text{AGGREGATE}\left(\{\mathbf{x}'_u, \forall u \in \mathcal{N}(v)\}\right)\right]$.

Since AGGREGATE($\cdot$) is linear, we have:

$$\mathbb{E}\left[\widehat{\mathbf{h}}_{\mathcal{N}(v)}\right] = \text{AGGREGATE}\left(\{\mathbb{E}\left[\mathbf{x}'_u\right], \forall u \in \mathcal{N}(v)\}\right).$$

Considering $\boldsymbol{\sigma} = 0$, we have $\mathbb{E}[\mathbf{x}'_u] = \mathbf{x}_u$ and hence have:

$$\mathbb{E}\left[\widehat{\mathbf{h}}_{\mathcal{N}(v)}\right] = \text{AGGREGATE}\left(\{\mathbf{x}_u, \forall u \in \mathcal{N}(v)\}\right) = \mathbf{h}_{\mathcal{N}(v)}.$$

$\square$

---

[3] Without loss of generality, this paper assumes that each dimension of the input data is normalized to the range $[-1, 1]$.

## B.2. Proof for Theorem 3

*Proof.* For any node $u \in \mathcal{V}$ and dimension $i \in \{1, 2, \ldots, d\}$, following the perturbation and calibration stages, we determine that $x'_{u,i} \in [-B, B]$, where $B$ is a finite boundary. We define $X = x'_{u,i} - x_{u,i}$. Given $x_{u,i} \in [-1, 1]$, we have: $|X| \leq B+1$, and through the calibration stage, if $\boldsymbol{\sigma} = 0$, we know that $\mathbb{E}[X] = \mathbb{E}[x'_{u,i}] - x_{u,i} = 0$.

Assuming the mean aggregator is employed, for any node $v \in \mathcal{V}$ and any dimension $i \in \{1, 2, \ldots, d\}$, we have:

$$(\mathbf{h}_{\mathcal{N}(v)})_i = \frac{1}{|\mathcal{N}(v)|} \sum_{u \in \mathcal{N}(v)} x_{u,i}, \quad (\widehat{\mathbf{h}}_{\mathcal{N}(v)})_i = \frac{1}{|\mathcal{N}(v)|} \sum_{u \in \mathcal{N}(v)} x'_{u,i}. \tag{20}$$

Thus, based on the Bernstein inequality, we have:

$$\Pr\left[\left|(\widehat{\mathbf{h}}_{\mathcal{N}(v)})_i - (\mathbf{h}_{\mathcal{N}(v)})_i\right| \geq \lambda\right] = \Pr\left[\left|\sum_{u \in \mathcal{N}(v)} (x'_{u,i} - x_{u,i})\right| \geq \lambda |\mathcal{N}(v)|\right] \tag{21}$$

$$\leq 2 \cdot \exp\left\{-\frac{\lambda^2 |\mathcal{N}(v)|}{\frac{2}{|\mathcal{N}(v)|} \sum_{u \in \mathcal{N}(v)} Var[X] + \frac{2}{3}\lambda(B+1)}\right\} = 2 \cdot \exp\left\{-\frac{\lambda^2 |\mathcal{N}(v)|}{2Var[x'_{u,i}] + \frac{2}{3}\lambda(B+1)}\right\}. \tag{22}$$

Considering Eq. (15) and Eq. (18), we have:

$$Var[x'_{u,i}] = \mathcal{O}(\frac{md}{\epsilon^2}). \tag{23}$$

Considering Eq. (16) and Eq. (19), we have:

$$B = \mathcal{O}(\frac{d}{\epsilon}). \tag{24}$$

Substituting the Eq. (23) and Eq. (24) into Eq. (22), we obtain:

$$\Pr\left[\left|(\widehat{\mathbf{h}}_{\mathcal{N}(v)})_i - (\mathbf{h}_{\mathcal{N}(v)})_i\right| \geq \lambda\right] \leq 2 \cdot \exp\left\{-\frac{\lambda^2 |\mathcal{N}(v)|}{\mathcal{O}(\frac{md}{\epsilon^2}) + \lambda\mathcal{O}(\frac{d}{\epsilon})}\right\}. \tag{25}$$

By applying the union bound, we have:

$$\Pr\left[\max_{i \in \{1,\ldots,d\}} \left|(\widehat{\mathbf{h}}_{\mathcal{N}(v)})_i - (\mathbf{h}_{\mathcal{N}(v)})_i\right| \geq \lambda\right] = \bigcup_{i=1}^{d} \Pr\left[\left|(\widehat{\mathbf{h}}_{\mathcal{N}(v)})_i - (\mathbf{h}_{\mathcal{N}(v)})_i\right| \geq \lambda\right] \tag{26}$$

$$\leq \sum_{i=1}^{d} \Pr\left[\left|(\widehat{\mathbf{h}}_{\mathcal{N}(v)})_i - (\mathbf{h}_{\mathcal{N}(v)})_i\right| \geq \lambda\right] = 2d \cdot \exp\left\{-\frac{\lambda^2 |\mathcal{N}(v)|}{\mathcal{O}(\frac{md}{\epsilon^2}) + \lambda\mathcal{O}(\frac{d}{\epsilon})}\right\}. \tag{27}$$

To ensure that $\max_{i \in \{1,\ldots,d\}} \left|(\widehat{\mathbf{h}}_{\mathcal{N}(v)})_i - (\mathbf{h}_{\mathcal{N}(v)})_i\right| < \lambda$ holds with at least $1 - \delta$ probability, it is sufficient to set $\delta = 2d \cdot \exp\left\{-\frac{\lambda^2 |\mathcal{N}(v)|}{\mathcal{O}(\frac{md}{\epsilon^2}) + \lambda\mathcal{O}(\frac{d}{\epsilon})}\right\}$. Solving the above for $\lambda$, we get: $\lambda = \mathcal{O}(\sqrt{d\log(d/\delta)}/(\epsilon\sqrt{|\mathcal{N}(v)|}))$. $\qquad\square$

## B.3. Proof for Theorem 4

*Proof.* Considering that

$$\Upsilon(\widehat{\mathbf{h}}) = \frac{1}{|\mathcal{V}|} \sum_{i \in \mathcal{V}} \sum_{j \in \mathcal{N}(v_i)} \left\|\widehat{\mathbf{h}}_i^k - \widehat{\mathbf{h}}_j^k\right\|_2^2 = \frac{1}{|\mathcal{V}|} \sum_{i \in \mathcal{V}} \sum_{j \in \mathcal{N}(v_i)} \left(\left\|\mathbf{h}_i^k - \mathbf{h}_j^k\right\|_2^2 + \underbrace{\left\|\eta_i^k - \eta_j^k\right\|_2^2}_{\text{noise signal}}\right) \tag{28}$$

and

$$\Phi_K = \lim_{K \to \infty} \frac{\sum_{k=1}^{K} \mathcal{E}_{\text{HOA}}^k}{\sum_{k=1}^{K} \mathcal{E}_{\text{SKA}}^k} = \lim_{K \to \infty} \frac{\sum_{k=1}^{K} \left(\frac{1}{|\mathcal{V}|} \sum_{i \in \mathcal{V}} \sum_{j \in \mathcal{N}(v_i)} |\widehat{\mathbf{h}}_{1i}^k - \widehat{\mathbf{h}}_{1j}^k|_2^2\right)}{\sum_{k=1}^{K} \left(\frac{1}{|\mathcal{V}|} \sum_{i \in \mathcal{V}} \sum_{j \in \mathcal{N}(v_i)} |\widehat{\mathbf{h}}_{2i}^k - \widehat{\mathbf{h}}_{2j}^k|_2^2\right)}, \tag{29}$$

although both HOA and SKA introduce the noise term $\eta^k$ in each iteration, the key difference lies in how they handle the propagation and amplification of noise. As explained in the main paper, HOA employs a personalized aggregation strategy, allowing it to better control the noise amplification effect in subsequent layers. Specifically, by assigning smaller aggregation weights to the neighbors in more distant levels, HOA can mitigate the cumulative impact of noise. Consequently, as $K$ (the number of aggregation steps) approaches infinity, the effect of noise diminishes more effectively in HOA, leading to the condition $\Phi_K = 0$.

$\square$

## B.4. Proof for Theorem 5

*Proof.* For the $L_1$-regularization problem addressed in this paper, we employ the *proximal gradient descent* (PGD) (Nitanda, 2014; Li & Lin, 2015; Duan et al., 2022) method for its solution. First, we obtain the derivative of $\mathcal{L}_1(\boldsymbol{w})$:

$$\nabla \mathcal{L}_1(\boldsymbol{w}) = \frac{1}{|\mathcal{N}(v)|} \sum_{u \in \mathcal{N}(v)} (\boldsymbol{w} - \mathbf{x}'_u) = \boldsymbol{w} - \frac{1}{|\mathcal{N}(v)|} \sum_{u \in \mathcal{N}(v)} \mathbf{x}'_u = \boldsymbol{w} - \widehat{\mathbf{h}}_v. \tag{30}$$

Then, the derivative of $\nabla \mathcal{L}_1(\boldsymbol{w})$ is $\frac{\mathrm{d}\nabla \mathcal{L}_1(\boldsymbol{w})}{\mathrm{d}\boldsymbol{w}} = 1$. According to *Cauchy mean value theorem*, we have:

$$\|\nabla \mathcal{L}_1(\boldsymbol{w}) - \nabla \mathcal{L}_1(\boldsymbol{w}_k)\|_2^2 \leq \|\boldsymbol{w} - \boldsymbol{w}_k\|_2^2, \tag{31}$$

where $\boldsymbol{w}_k$ represents the result of the $k$-th iteration.
We can approximate $\mathcal{L}_1(\boldsymbol{w})$ around $\boldsymbol{w}_k$ using *second-order Taylor expansion*, given by:

$$\mathcal{L}_1(\boldsymbol{w}) \simeq \mathcal{L}_1(\boldsymbol{w}_k) + \langle \nabla \mathcal{L}_1(\boldsymbol{w}_k), \boldsymbol{w} - \boldsymbol{w}_k \rangle + \frac{1}{2} \|\boldsymbol{w} - \boldsymbol{w}_k\|^2$$

$$= \frac{1}{2} \|\boldsymbol{w} - (\boldsymbol{w}_k - \nabla \mathcal{L}_1(\boldsymbol{w}_k))\|_2^2 + \mathrm{const},$$

where $\mathrm{const}$ is a constant independent of $\boldsymbol{w}$ and $\langle \cdot, \cdot \rangle$ denotes the inner product. The minimum of the above equation is obtained at $\boldsymbol{w}_{k+1}$ as follows:

$$\boldsymbol{w}_{k+1} = \boldsymbol{w}_k - \nabla \mathcal{L}_1(\boldsymbol{w}_k). \tag{32}$$

We then introduce the $L_1$-regularization term in the iteration:

$$\boldsymbol{w}_{k+1} = \arg\min_{\boldsymbol{w}} \frac{1}{2} \|\boldsymbol{w} - (\boldsymbol{w}_k - \nabla \mathcal{L}_1(\boldsymbol{w}_k))\|_2^2 + \mu_1 \|\boldsymbol{w}\|_1. \tag{33}$$

Let $\boldsymbol{w}_i$ denote the $i$-th dimension of $\boldsymbol{w}$. Since each dimension is independent, we have:

$$(\boldsymbol{w}_{k+1})_i = \arg\min_{\boldsymbol{w}_i} \frac{1}{2} \|\boldsymbol{w}_i - ((\boldsymbol{w}_k)_i - \nabla \mathcal{L}((\boldsymbol{w}_k)_i))\|_2^2 + \mu_1 \|(\boldsymbol{w})_i\|_1. \tag{34}$$

Let $\boldsymbol{z}_i = (\boldsymbol{w}_k)_i - \nabla \mathcal{L}_1((\boldsymbol{w}_k)_i) = (\boldsymbol{w}_k)_i - \left((\boldsymbol{w}_k)_i - (\widehat{\mathbf{h}}_v)_i\right) = (\widehat{\mathbf{h}}_v)_i$. Computing $(\boldsymbol{w}_{k+1})_i$ depends on whether $\boldsymbol{w}_i$ is positive, negative or 0. Below, we compute it in the following cases:

- If $\boldsymbol{w}_i > 0$, by setting the gradient $\boldsymbol{w}_i - \boldsymbol{z}_i + \mu_1$ of Eq. (34) equal to zero, we obtain $(\boldsymbol{w}_{k+1})_i = \boldsymbol{z}_i - \mu_1 > 0$, hence $\boldsymbol{z}_i > \mu_1$.

- If $\boldsymbol{w}^i < 0$, similarly, we obtain $(\boldsymbol{w}_{k+1})_i = \boldsymbol{z}_i + \mu_1 < 0$, hence $\boldsymbol{z}_i < -\mu_1$.

- If $\boldsymbol{w}_i = 0$, we obtain $(\boldsymbol{w}_{k+1})_i = 0$, in which case $|\boldsymbol{z}_i| \leq \mu_1$.

Consequently, we have:

$$(\boldsymbol{w}_{k+1})_i = \mathrm{sign}(\boldsymbol{z}_i) \cdot \max(|\boldsymbol{z}_i| - \mu_1, 0), \tag{35}$$

where $\mathrm{sign}$ denotes the sign function, which takes 1 if $(\widehat{\mathbf{h}}_v)_i > 0$, 0 if $(\widehat{\mathbf{h}}_v)_i = 0$, and -1 if $(\widehat{\mathbf{h}}_v)_i < 0$. Applying $(\boldsymbol{w}_{k+1})_i = (\widetilde{\mathbf{h}}_v)_i$ and $\boldsymbol{z}_i = (\widehat{\mathbf{h}}_v)_i$.

$$(\widetilde{\mathbf{h}}_v)_i = \mathrm{sign}\left((\widehat{\mathbf{h}}_v)_i\right) \cdot \max\left(|(\widehat{\mathbf{h}}_v)_i| - \mu_1, 0\right), \tag{36}$$

The optimal value for $\mu_1$ is $\tau_1 B/\bar{d}^K$, where $\tau_1 \in (0,1)$, with $B$ as the boundary of the perturbed node features, $\bar{d}$ as the approximate average degree of the graph, and $K$ as the step parameter of the HOA layer. Here, $\bar{d}^K$ approximates the number of neighbors, as a larger number of neighbors tends to smooth out the feature values, allowing for a relatively smaller regularization parameter $\mu_1$. This formulation reflects how the regularization strength can adjust according to the feature magnitude and the neighborhood structure. $\qquad\square$

### B.5. Proof for Theorem 6

*Proof.* Under the N-H architecture, the NFR layer aims to enhance utility through efficient feature selection of the perturbed node features $\mathbf{x}'$ using $L_1$-regularization. The objective function $\mathcal{L}_2$ is defined as follows:

$$\mathcal{L}_2(\mathbf{x}) = \frac{1}{2}\left\|\mathbf{x}' - \mathbf{x}\right\|_2^2 + \mu_2 \left\|\mathbf{x}\right\|_1. \tag{37}$$

Our goal is to minimize the above loss function. The proof procedure is similar to that of Theorem 5 and is not repeated. $\quad\square$

## C. More Details of N-H Architecture

In this section, We provide more details on the N-H architecture of UPGNET, where the perturbed node features $\mathbf{x}'$ are sequentially enhanced by first passing through the NFR layer followed by the HOA layer.

**Node Feature Regularization.** Under the N-H architecture, the NFR layer specifically aims to enhance utility through efficient feature selection of the perturbed node features $\mathbf{x}'$ directly using $L_1$-regularization. The objective function $\mathcal{L}_2$ is formalized as follows:

$$\mathcal{L}_2(\mathbf{x}) = \frac{1}{2}\left\|\mathbf{x}' - \mathbf{x}\right\|_2^2 + \mu_2 \left\|\mathbf{x}\right\|_1. \tag{38}$$

Our goal is to minimize the above loss function. Thm. 7 derives this regularization problem. According to Thm. 7, we conclude that Eq. (39) can efficiently achieve feature selection for $\mathbf{x}'_v$, thereby enhancing the utility of private graph learning.

**Theorem 7.** *For any node $v$ and any $i \in \{1, \dots, d\}$, $(\widetilde{\mathbf{x}}_v)_i$ in the following equation can efficiently achieve feature selection:*

$$(\widetilde{\mathbf{x}}_v)_i = \text{sign}\left((\mathbf{x}'_v)_i\right) \cdot \max\left(|(\mathbf{x}'_v)_i| - \mu_2, 0\right), \tag{39}$$

*where the optimal value for $\mu_1$ is $\tau_2 B$, where $\tau_2 \in (0,1)$, with $B$ as the boundary of the perturbed node features.*

**Higher-Order Aggregator.** After utility enhancement through the NFR layer, additional utility enhancement is achieved by executing Alg. 1 on the enhanced feature vector $\widetilde{\mathbf{x}}_v$ through the HOA layer.

## D. Privacy Analysis

The piecewise mechanism (PM) (Wang et al., 2019a; Pei et al., 2023) and multi-bit mechanism (MBM) (Du et al., 2021; Sajadmanesh & Gatica-Perez, 2021; Lin et al., 2022; Jin & Chen, 2022) satisfy $\epsilon$-local differential privacy for each node. The entire training process remains LDP-compliant due to the robustness of differential privacy against *post-processing* (Dwork et al., 2014) (Thm. 8). Moreover, any subsequent prediction is bounded by the post-processing theorem (Dwork et al., 2014), since the LDP is applied only once to the private data. This ensures that LDP holds for all nodes throughout the process.

**Definition 8** (Post-processing). *If an algorithm $\mathcal{A}(\cdot)$ satisfies $\epsilon$-local differential privacy, then any further processing of its output by another algorithm $\mathcal{B}(\cdot)$ (i.e., process $\mathcal{B}(\mathcal{A}(\cdot))$) also maintains $\epsilon$-local differential privacy.*

## E. Complexity Analysis

The computational complexity of UPGNET mainly arises from its two key components: the HOA layer and the NFR layer. The HOA layer performs $K$ steps of high-order neighborhood aggregation, where each step aggregating feature vectors from neighboring nodes. This process has a time complexity of $\mathcal{O}(|\mathcal{E}| \cdot d)$, where $|\mathcal{E}|$ is the number of edges and $d$ is the feature dimension, resulting in a total complexity of $\mathcal{O}(K \cdot |\mathcal{E}| \cdot d)$. The NFR layer applies lightweight element-wise transformations to each node's feature vector, contributing a complexity of $\mathcal{O}(|\mathcal{V}| \cdot d)$, where $|\mathcal{V}|$ is the number of nodes. Combining these, the overall computational complexity of UPGNET is $\mathcal{O}(K \cdot |\mathcal{E}| \cdot d + |\mathcal{V}| \cdot d)$, scaling linearly with graph size and feature dimensionality. This ensures that UPGNET remains both practical and scalable for large-scale graphs with

high-dimensional data. Furthermore, compared to other methods (Solitude (Lin et al., 2022) and LPGNN (Sajadmanesh & Gatica-Perez, 2021)), the computational complexity of UPGNET introduces only an additional factor $|\mathcal{V}| \cdot d$. This factor is linear with respect to the number of nodes and the feature dimension, making it highly efficient in practice. Moreover, graph pruning (Yu et al., 2022; Liu et al., 2023) or parallelization techniques, such as GPU-based sparse matrix operations (Lee et al., 2020), can be applied to further reduce computational overhead.

## F. Additional Details of Experiments

### F.1. More Parameter Design

To evaluate the performance of UPGNET, we use three state-of-the-art GNN architectures, graph convolutional networks (GCN) (Kipf & Welling, 2017) GraphSAGE (Hamilton et al., 2017) and graph attention networks (GAT) (Velickovic et al., 2018), as the backbone models. All GNN models have two graph convolutional layers, each with a hidden dimension of size 16 and a SeLU activation function (Klambauer et al., 2017) followed by dropout. The GAT model has four attention heads. To obtain the best hyperparameters, we use grid search for selection: both learning rate and weight decay are chosen from $\{10^{-4}, 10^{-3}, 10^{-2}, 10^{-1}\}$, and dropout is chosen from $\{10^{-4}, 10^{-3}, 10^{-2}, 10^{-1}\}$. The HOA's step parameter is denoted by $K$. Based on the selected best hyperparameters, the best $K$ within $\{0, 2, 4, 8, 16, 32, 64\}$ for all $\epsilon \in \{0.01, 0.1, 1, 2, 3\}$. The parameters $\tau_1, \tau_2$ of the NFR layer belong to $\{0.1, 0.3, 0.5, 0.7, 0.9\}$. We use the Adam optimizer (Kingma & Ba, 2014) for all models. Without specification, we report the average results based on the best of each parameter.

### F.2. More Descriptions about the GNN Model

- **GCN** (Kipf & Welling, 2017) applies spectral graph convolution by propagating node features using the Laplacian matrix. The core update rule is: $H^{(l+1)} = \sigma(\tilde{D}^{-1/2}\tilde{A}\tilde{D}^{-1/2}H^{(l)}W^{(l)})$, where $\tilde{A} = A + I$ is the adjacency matrix with self-loops, $\tilde{D}$ is its degree matrix, $W^{(l)}$ is the trainable weight matrix, and $\sigma$ is a activation function.

- **GraphSAGE** (Hamilton et al., 2017) samples a fixed number of neighbors and aggregates their features, making it scalable for large graphs. The general update rule is: $h_v^{(l+1)} = \sigma(W^{(l)} \cdot \text{AGGREGATE}(\{h_u^{(l)}, \forall u \in \mathcal{N}(v)\}))$, where $\text{AGGREGATE}(\cdot)$ represents the aggregation operation.

- **GAT** (Hamilton et al., 2017) employs self-attention to dynamically assign importance to neighbors. The update rule is: $h_v^{(l+1)} = \sigma\left(\sum_{u \in \mathcal{N}(v)} \alpha_{vu} W^{(l)} h_u^{(l)}\right)$, where the attention coefficient $\alpha_{vu}$ is computed as:

$$\alpha_{vu} = \frac{\exp(\text{LeakyReLU}(a^T[W^{(l)}h_v^{(l)}\|W^{(l)}h_u^{(l)}]))}{\sum_{k \in \mathcal{N}(v)} \exp(\text{LeakyReLU}(a^T[W^{(l)}h_v^{(l)}\|W^{(l)}h_k^{(l)}]))}. \tag{40}$$

Here, $\alpha$ is a learnable vector, and $\|$ denotes concatenation.

### F.3. Impact of Average Node Degree of Dataset on HOA

As shown in Table 1, the social network datasets, Facebook (Rozemberczki et al., 2021) and LastFM (Rozemberczki & Sarkar, 2020), exhibit a higher average node degree (*Deg.*) compared to citation networks (Cora (Yang et al., 2016) and CiteSeer (Yang et al., 2016)). For Facebook and LastFM, as illustrated in Fig. 6, the accuracy leveled off after $K = 1$, despite further increases in $K$. This suggests that for social networks with higher average node degrees, the initial aggregation step provides a sufficient representation of node information. In contrast, for lower-degree citation networks Cora and CiteSeer, the accuracy continues to improve until $K = 64$ after $K = 1$. This indicates that in low-degree datasets, HOA requires more steps to aggregate enough neighboring nodes for better accuracy.

### F.4. Scalability on Heterophilic Graphs

When applied to heterophilic graphs (*e.g.*, Flickr (Huang et al., 2017) and Reddit (Hamilton et al., 2017); see Table 3 for detailed statistics), UPGNET continues to demonstrate superior utility compared to other baselines, as shown in Fig. 7. The HOA layer, by preserving Dirichlet energy and enabling personalized aggregation, effectively calibrates noise in heterophilic graphs, as evidenced by the experiments in Fig. 8.

### F.5. Ablation Study on HOA layer

*Table 3.* Statistics of heterophilic graph datasets.

| DATASET | #CLASSES | #NODES | #EDGES | #FEATURES |
|---|---|---|---|---|
| FLICKR | 7 | 89,250 | 899,756 | 500 |
| REDDIT | 41 | 232,965 | 114,615,892 | 602 |

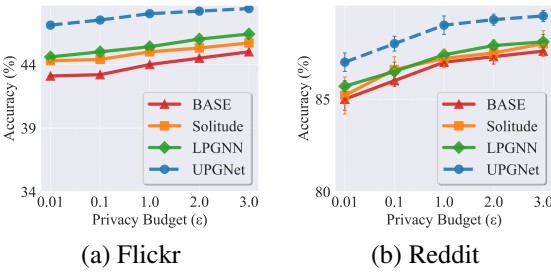

(a) Flickr          (b) Reddit

*Figure 7.* Performance comparison of UPGNET and other baselines on Flickr and Reddit. X-axis represents $\epsilon$ and y-axis represents test accuracy (%). UPGNET exhibits superior performance compared to other baselines.

(a) Flickr          (b) Reddit

*Figure 8.* Effect of HOA vs. SKA on graph learning performance across various steps $K \in \{2, 4, 8, 16, 32, 64\}$. HOA demonstrates its superior denoising capability on heterophilic datasets (Flickr and Reddit).

Fig. 9 presents an additional experiment demonstrating its superiority in private graph learning compared to residual connections (RC) (Liu et al., 2021). RC primarily addresses vanishing gradients but remains ineffective in suppressing LDP noise. In contrast, HOA mitigates oversmoothing by preserving Dirichlet energy, thereby effectively calibrating noise.

### F.6. NFR vs. other Regularization Approaches

As stated in Thm. 3, a key aspect of noise calibration in LDPGNN lies in reducing the effective feature dimensions. NFR employs L1 regularization, which directly facilitates feature selection and enhances fine-grained noise calibration. In contrast, Dropout (Baldi & Sadowski, 2013) and Group Lasso (Meier et al., 2008) fail to achieve precise feature selection tailored for noise calibration. Dropout introduces randomness by stochastically deactivating neurons, leading to instability in feature selection, while Group Lasso enforces sparsity at a predefined group level, requiring carefully designed group definitions that may not align with optimal noise calibration. These limitations reduce their effectiveness in mitigating noise impact. Empirical results (Table 4) further validate that NFR outperforms Dropout and Group Lasso in preserving learning utility.

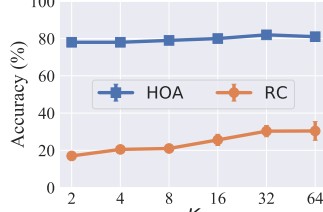

*Figure 9.* Comparison of HOA and Residual Connection (RC) (Dataset: Cora, $\epsilon = 0.01$). HOA demonstrates significantly better classification accuracy in private graph learning compared to RC.

*Table 4.* Comparison of our NFR (without HOA) with dropout and group Lasso ($\epsilon = 0.01$, GCN). The values in the table represent accuracy (%). NFR demonstrates significantly better learning utility compared to other sparsity-inducing techniques.

| BASELINE | CORA | CITESEER | LASTFM | FACEBOOK |
|---|---|---|---|---|
| DROPOUT | 63.4 | 52.7 | 61.3 | 77.6 |
| GROUP LASSO | 57.6 | 50.5 | 57.1 | 78.9 |
| **OURS** | **71.3** | **57.2** | **67.1** | **84.9** |

