# OpenReview forum: "Going Deeper into Locally Differentially Private Graph Neural Networks"
_ICML.cc/2025/Conference — ICML 2025 oral_

### Official Review · Reviewer_jWSr · 2025-03-05

**Overall Recommendation:** 4

**Summary:**

This paper presents UPGNet, a utility-enhanced framework for locally private graph learning. The main contribution is a three-stage pipeline that generalizes local differential privacy protocols for perturbing node features, aiming to balance privacy preservation with improved learning utility. The authors identify two key factors influencing utility: feature dimension and neighborhood size. To address these, they propose two novel components: the Node Feature Regularization (NFR) layer, which reduces the effective feature dimension, and the High-Order Aggregator (HOA) layer, which mitigates over-smoothing by expanding the effective neighborhood size. The theoretical analysis and experimental results validate the effectiveness of the proposed framework in achieving a balance between privacy and utility in graph learning tasks.

**Claims And Evidence:**

Yes

**Essential References Not Discussed:**

The citations and related work in the paper are comprehensive.

**Experimental Designs Or Analyses:**

I reviewed the experimental designs and analyses presented in the paper, particularly those evaluating the performance of the proposed UPGNet framework. The experimental setup includes comprehensive testing on four benchmark datasets and compares UPGNet against several baselines. The experimental results validate the effectiveness of the proposed framework in achieving a balance between privacy and utility in graph learning tasks.

**Methods And Evaluation Criteria:**

Yes

**Other Comments Or Suggestions:**

see the weaknesses

**Other Strengths And Weaknesses:**

Strengths:

(i)Originality and Innovation: The problem of this paper is interesting and significant, and the method proposed by the authors significantly improves the utility of privacy-preserving graph learning without compromising privacy.

(ii)Clear Theoretical and Experimental Support: The paper is well-structured, with a solid theoretical foundation to justify the proposed methods.

(iii)Practical Relevance: The proposed methods and their demonstrated effectiveness in privacy-preserving settings could have broad applications, making the research practically significant.

Weaknesses:

(i)Provides more description of the node categorization task. This paper conducts experiments based on the node classification task and verifies the effectiveness of the UPGNet method in this paper in terms of utility enhancement compared to other methods. Providing more specific instructions on the node classification task is necessary.

(ii)The theorem to be referenced should have been presented earlier in the text.

(iii)Equation (2) in the paper uses the superscript $k$ to indicate the layer, while Equation (7) uses the superscript $(k)$ in a slightly different notation. This inconsistency in formalization can confuse readers, making it harder to follow the mathematical derivations. It is suggested that the authors review the formalization of these equations and ensure that the notation for layers is consistent throughout the paper.

**Questions For Authors:**

(i)How does the HOA layer in the paper differ from multi-hop graph structures?

(ii)How is the effectiveness of NFR in improving utility validated in the experiments?

**Relation To Broader Scientific Literature:**

The key contributions of this paper are closely related to the broader scientific literature on privacy-preserving machine learning, particularly in the context of Graph Neural Networks (GNNs) and local differential privacy (LDP). The paper's key contributions pushes forward both the theoretical and practical aspects of applying LDP to GNNs.

**Theoretical Claims:**

Yes, I checked

---

> ### Author Rebuttal · Authors · 2025-03-31
>
> We sincerely appreciate your valuable comments and suggestions, which have significantly contributed to improving the quality of our paper. Detailed responses to each comment are provided below.
>
>
> **Q1: Provides more description of the node classification task. This paper conducts experiments based on the node classification task and verifies the effectiveness of the UPGNet method in this paper in terms of utility enhancement compared to other methods. Providing more specific instructions on the node classification task is necessary.**
>
> **R1:** Thank you for your valuable suggestion. A more detailed description of the node classification task, along with its formal definition, will be included in the revised manuscript.
>
> The node classification task [1][2] is a fundamental problem in graph learning, where the objective is to predict the labels of nodes based on the graph structure and available node features. Given a graph $\mathcal{G} = (\mathcal{V}, \mathcal{E})$, where $\mathcal{V}$ is the set of nodes and $\mathcal{E}$ is the set of edges, each node $v_i \in \mathcal{V}$ is associated with a feature vector $\mathbf{x}_i \in \mathbb{R}^d$ and may have a corresponding label $y_i$ from a predefined label set $\mathcal{Y}$. The goal is to learn a function $f: \mathcal{V} \to \mathcal{Y}$ that assigns labels to unlabeled nodes based on their attributes and graph connectivity.
>
> > [1] Kipf, Thomas N., and Max Welling. "Semi-Supervised Classification with Graph Convolutional Networks." ICLR. 2017.\
>  [2] Bhagat, Smriti, Graham Cormode, and S. Muthukrishnan. "Node classification in social networks." Social network data analytics (2011): 115-148.
>
> **Q2: The theorem to be referenced should have been presented earlier in the text.**
>
> **R2:** Thank you for your suggestion. The manuscript will be revised to ensure that the referenced theorem appears earlier in the text, improving the logical flow and readability.
>
> **Q3: Equation (2) in the paper uses the superscript $k$ to indicate the layer, while Equation (7) uses the superscript $(k)$ in a slightly different notation. This inconsistency in formalization can confuse readers, making it harder to follow the mathematical derivations. It is suggested that the authors review the formalization of these equations and ensure that the notation for layers is consistent throughout the paper.**
>
> **R3:** Thank you for pointing out this inconsistency in our notation. The notation for layer indices has been carefully reviewed and unified throughout the paper to improve clarity and consistency. Specifically, we now consistently use the superscript [$\cdot^k$] across all equations to denote the $k$-th layer in our model.
>
> **Q4: How does the HOA layer in the paper differ from multi-hop graph structures?**
>
> **R4:** Thank you for your question. The key difference between the High-Order Aggregator (HOA) and traditional multi-hop graph structures lies in the way information is aggregated across different neighborhood scales. While multi-hop methods explicitly expand the receptive field to fixed-hop neighborhoods, HOA dynamically balances information across different hops, ensuring that the energy ratio $\Phi_K$ (Equation 8 in Section 3.2.2) approaches 0 as $K \to \infty$, which reduces over-smoothing. Unlike standard multi-hop aggregation, which applies uniform or predefined weight decay, HOA assigns personalized weightings, prioritizing closer neighbors to enhance noise calibration. This design prevents excessive noise propagation from distant nodes while still leveraging high-order information adaptively. These differences make HOA more effective in maintaining utility under noisy conditions. For more details, please refer to Section 3.2.2 on HOA and Section 4.4 on HOA ablation experiments in the paper.
>
> **Q5: How is the effectiveness of NFR in improving utility validated in the experiments?**
>
> **R5:** Thank you for your question. In Section 4.3, a detailed ablation study is conducted to validate the effectiveness of the Node Feature Regularizer (NFR) layer in improving utility. Specifically, the node classification accuracy of the piecewise mechanism (PM) and multi-bit mechanism (MBM) with and without the NFR layer is compared across different datasets and privacy budgets.  As shown in Table 2 in the paper ($\star$ in the table represents the integration of NFR), incorporating NFR layer consistently enhances accuracy, with more significant improvements under smaller privacy budgets. For example, in the Cora dataset, MBM$^\star$ improves accuracy by 7% over MBM at $\epsilon=0.01$, while the improvement is 2.6% at $\epsilon=1.0$. This demonstrates that NFR effectively mitigates noise impact, especially under strong privacy constraints. Additional results are provided in Appendix D.5.

---

### Official Review · Reviewer_XrJc · 2025-03-07

**Overall Recommendation:** 4

**Summary:**

This study introduces the UPGNET framework, which aims to protect user privacy through Local Differential Privacy (LDP) while enhancing the learning utility of graph neural networks. It innovatively proposes the High-Order Aggregator and Node Feature Regularization layers to optimize feature dimensions and neighborhood size. Experimental results demonstrate that UPGNET outperforms existing methods on real-world graph datasets. This work holds significant implications for private machine learning, particularly in the context of privacy-preserving graph neural networks.

**Claims And Evidence:**

The claims made in the study are supported by clear and convincing evidence. The paper provides a solid theoretical foundation for the proposed methods, such as the analysis of key factors impacting the utility of privacy-preserving graph learning. Additionally, extensive experiments on benchmark datasets validate the effectiveness of the proposed framework UPGNET in terms of both privacy preservation and learning utility. The performance comparisons with baseline methods like LPGNN and Solitude further strengthen the claims.

**Essential References Not Discussed:**

There are no essential references that have been overlooked in this paper.

**Experimental Designs Or Analyses:**

I checked the experimental sections of this paper, including the experimental setup, performance validation of UPGNET, ablation study of NFR and HOA, architecture comparison and empirical privacy attack defense experiments. The experimental validation of this study is adequate and detailed.

**Methods And Evaluation Criteria:**

The proposed methods and evaluation criteria, including the benchmark datasets (Cora, CiteSeer, LastFM, Facebook), are well-suited to the problem at hand.

**Other Comments Or Suggestions:**

Suggestion 1: Conducting additional experiments using different GNN models, such as GAT, would provide a more comprehensive evaluation of the framework's performance.
Suggestion 2: Providing an explanation of what the ↑ symbol represents would enhance the reader’s understanding of the data presented.
Suggestion 3: Add the necessary references.

**Other Strengths And Weaknesses:**

Strengths
1. this paper addresses significant challenges in privacy-preserving gnn, with a special focus on utility loss due to local differential privacy perturbations.
2. this paper validates the effectiveness of UPGNET for utility enhancement and privacy preservation through detailed theoretical analyses and extensive experiments.
3. this paper is well organized and well written.

Weaknesses
1. Adding more experiments under the GNN model, e.g. GAT. While the paper provides valuable insights into the GNN model's performance, adding experiments specifically focused on GAT would provide a more comprehensive comparison of different GNN architectures.
2. In the experiment section, the use of the ↑ symbols in Table 1 should be clearly defined. Providing an explanation of what the ↑ symbol represents would enhance the reader’s understanding of the data presented.
3. There are places in the article where necessary citations should be added, such as line 176 in the right-hand column on page 4, where citations should be added for proximal gradient descent (PGD).

**Questions For Authors:**

1.	Why is it required to select m dimensions for perturbation in d-dimensional node features under LDP?
2.	Can the method proposed in the paper be applied to more GNN frameworks such as GAT?
3.	Why use node classification accuracy as a metric?

**Relation To Broader Scientific Literature:**

This paper addresses an important and timely issue in privacy preserving machine learning. The methods in the paper effectively enhance the utility of privacy graph learning.

**Theoretical Claims:**

I checked the proofs in this paper, including unbiased estimation, key factor analysis, feature sparsification analysis in NFR, etc., and these proofs are correct.

---

> ### Author Rebuttal · Authors · 2025-03-31
>
> We sincerely appreciate your valuable comments and suggestions, which have significantly contributed to improving the quality of our paper. Detailed responses to each comment are provided below.
>
> **Q1: Adding more experiments under the GNN model, e.g. GAT. While the paper provides valuable insights into the GNN model's performance, adding experiments specifically focused on GAT would provide a more comprehensive comparison of different GNN architectures.**
>
> **R1:** We appreciate the reviewer’s suggestion to include experiments with GAT to provide a more comprehensive comparison across different GNN architectures. These experiments have been conducted, and the results are presented in Appendix F.6. Specifically, Figure 6 in the original manuscript illustrates the performance of UPGNet compared to various baselines (BASE, Solitude, and LPGNN) under GAT with different privacy budgets $\epsilon \in$ {0.01, 0.1, 1.0, 2.0, 3.0}. The results demonstrate that UPGNet consistently outperforms these baselines across all privacy settings, further confirming its effectiveness in enhancing both utility and privacy preservation in graph learning. To improve clarity and ensure that readers do not overlook these results, the main text will be revised to explicitly highlight the presence of GAT-based experiments.
>
> **Q2: In the experiment section, the use of the ↑ symbols in Table 1 should be clearly defined. Providing an explanation of what the ↑ symbol represents would enhance the reader’s understanding of the data presented.**
>
> **R2:** Thank you for your valuable feedback. The ↑ symbols in Table 1 represent the utility improvement achieved by integrating the NFR layer, indicating the enhancement in performance when NFR is applied compared to when it is not. To ensure clarity for readers, an explanation of the ↑ symbol will be added to both the table caption and the surrounding text, making it clear that this symbol indicates the increase in utility after incorporating the NFR layer.
>
> **Q3: There are places in the article where necessary citations should be added, such as line 176 in the right-hand column on page 4, where citations should be added for proximal gradient descent (PGD).**
>
> **R3:** Thank you for the suggestion. The appropriate citation for proximal gradient descent (PGD) will be added in line 176 to ensure proper referencing.
>
> **Q4: Why is it required to select m dimensions for perturbation in d-dimensional node features under LDP?**
>
> **R4:** Thank you for your question. The selection of $m$ dimensions for perturbation in $d$-dimensional node features under LDP is a design choice aimed at controlling the trade-off between privacy and utility. By randomly selecting $m$ dimensions from the $d$ available dimensions, the mechanism focuses on perturbing only a subset of the features, thereby limiting the extent of noise injected. Each selected dimension is perturbed with a privacy budget of $\epsilon/m$, ensuring that the total privacy budget is allocated evenly across the perturbed dimensions. This strategy balances privacy preservation and the maintenance of data utility.
>
> **Q5: Can the method proposed in the paper be applied to more GNN frameworks such as GAT?**
>
> **R5:** Thank you for your question. As mentioned in Q2, experiments with GAT are presented in Appendix F.6 of the original manuscript. These results demonstrate that our approach is effective across different GNN architectures, confirming its generalizability and applicability to a variety of graph learning models.
>
> **Q6: Why use node classification accuracy as a metric?**
>
> **R6:** Thank you for your question. Node classification accuracy is used as a metric because it directly measures the effectiveness of graph-based models in learning meaningful representations of nodes, which is central to many graph learning tasks. This metric allows us to evaluate the trade-off between privacy preservation and utility. Additionally, node classification accuracy aligns with prior work (Sajadmanesh & Gatica-Perez, 2021; Lin et al., 2022) in the field, ensuring consistency and comparability of our results with established benchmarks. By using this metric, we maintain continuity with existing research while demonstrating the ability of our proposed methods.

---

### Official Review · Reviewer_V5sG · 2025-03-10

**Overall Recommendation:** 4

**Summary:**

This paper aims to enhance the utility of locally differential privacy graph learning. Its theoretical analysis derives two key factors affecting the estimation error, i.e., feature dimension and neighborhood size, and concludes that reducing the effective feature dimension and expanding the effective neighborhood size are conducive to enhancing the utility. Based on this conclusion, the paper proposes NFR and HOA layers to optimize the feature dimension and neighborhood scale. The generalization and effectiveness of UPGNet and its components are verified through theoretical analysis and experimental validation.

**Claims And Evidence:**

Yes, this paper confirms its claims through detailed theoretical analysis and experimental validation.

**Essential References Not Discussed:**

No.

**Experimental Designs Or Analyses:**

Yes, I have checked the soundness of the experimental design, validation and analysis of this paper.

**Methods And Evaluation Criteria:**

- The proposed method effectively enhances the utility of privacy graph learning.
- The baselines compared and the benchmark datasets employed are comprehensive and sensible.
- The adopted evaluation metrics are reasonable and aligned with previous work.

**Other Comments Or Suggestions:**

- Provide an analysis on the impact of graph density on the performance.
- Provides details of different GNN architectures. The model performance under different GNN architectures of GCN, GraphSAGE and GAT was validated and compared in the paper, and specific implementation details of the three different GNN architectures should be provided.
- Adjust charts for clarity.

**Other Strengths And Weaknesses:**

# Strengths:
- **Novel and important problem**. With the increasing use of GNNs in privacy-sensitive domains such as social networking and bioinformatics, the issue of data privacy has become critical. This paper hopes to enhance the model utility while utilizing local differential privacy to achieve privacy preservation, making it more practical and promising for real-world applications in sensitive data environments.
- **Theoretical Innovations**. This paper derives two key factors affecting the estimation error, i.e. feature dimension and neighborhood size, through theoretical analysis. It is concluded that reducing the effective feature dimension and enlarging the effective neighborhood size are conducive to improving the utility. Then, UPGNet integrates two layers, HOA and NFR, expands the effective neighborhood range, and applies L1 regularization for feature sparsification, which reduces the estimation error in the LDP setting. These analyses and components represent important theoretical innovations.
- **Sufficient experimental validation**.  This paper conducts detailed experiments on multiple datasets to illustrate the performance of UPGNet in various privacy budgets. The results show that the performance of UPGNet is stable and accurate, proving the practicality and robustness of the proposed framework.


# Weaknesses:
- **Provide an analysis on the impact of graph density on the performance**. The difference in graph density—whether a graph is sparse or dense—can play a crucial role in the performance of models, especially in tasks involving graph neural networks. The difference in the density of different graphs, does this have an impact on the model performance in this paper and what is the impact?
- Provides details of different GNN architectures. The paper has validated and compared model performance under different GNN architectures, including Graph Convolutional Networks (GCN), GraphSAGE, and Graph Attention Networks (GAT). However, it would be beneficial to offer more comprehensive implementation details for these architectures. This will help the reader to better understand how these models are applied and compared.
- **Adjust charts for clarity**. Adjust the vertical coordinate spacing in the graphs to improve clarity, e.g., Figure 9.

**Questions For Authors:**

- Can you provide details on different GNN architectures?
- Could the charts be adjusted for better clarity?
- How does the density of the datasets impact the performance?

**Relation To Broader Scientific Literature:**

With the increasing use of GNNs in privacy-sensitive domains such as social networking and bioinformatics, the issue of data privacy has become critical. This paper hopes to enhance the model utility while utilizing local differential privacy to achieve privacy preservation, making it more practical and promising for real-world applications in sensitive data environments.

**Theoretical Claims:**

Yes, I have checked the correctness of the five proofs of this paper.

---

> ### Author Rebuttal · Authors · 2025-03-31
>
> We sincerely appreciate your valuable comments and suggestions, which have significantly contributed to improving the quality of our paper. Detailed responses to each comment are provided below.
>
> **Q1: Provide an analysis on the impact of graph density on the performance. The difference in graph density—whether a graph is sparse or dense—can play a crucial role in the performance of models, especially in tasks involving graph neural networks. The difference in the density of different graphs, does this have an impact on the model performance in this paper and what is the impact?**
>
> **R1:** Thank you for the valuable suggestion. As detailed in Appendix F.6, our experiments indicate that graph density has a noticeable impact on the performance of the proposed method. The table below shows the average node degree (Avg. Deg.) of the four datasets used in the paper:
>
> | Dataset | Cora | CiteSeer | LastFM | Facebook |
> | ---- | ---- | ---- |---- |---- |
> |Avg. Deg.|	3.90 |	2.74 |	7.29|	15.21|
>
> Specifically, social network datasets like Facebook and LastFM, which exhibit higher average node degrees (higher graph density), show that accuracy levels off after just a few aggregation steps (e.g., K = 1). This is because the dense nature of these networks allows effective information aggregation early on, making additional steps less effective.
>
> In contrast, for sparser graphs like Cora and CiteSeer, the accuracy continues to improve with additional aggregation steps (up to K = 64), indicating that more steps are required to gather sufficient neighboring information for better node representation. These observations suggest that the performance of our method is influenced by the density of the graph, with denser graphs benefiting from fewer aggregation steps and sparser graphs requiring more steps for improved accuracy. For further details, please refer to Appendix F.6 in our paper.
>
> **Q2:  Provides details of different GNN architectures. The paper has validated and compared model performance under different GNN architectures, including Graph Convolutional Networks (GCN), GraphSAGE, and Graph Attention Networks (GAT). However, it would be beneficial to offer more comprehensive implementation details for these architectures. This will help the reader to better understand how these models are applied and compared.**
>
> **R2:** Thank you for the valuable suggestion. Appendix F of the paper provides details on the configurations of three different GNN models. To clarify these configurations further, the following elaboration is provided: All GNN models (GCN, GraphSAGE, and GAT) consist of two graph convolutional layers, each with a hidden dimension of 16 and a SeLU activation function, followed by dropout layer. The GAT model employs four attention heads.
>
> Additionally, we have added implementation details of these three GNN models to enhance readers' understanding of their application and comparison, as follows:
>
> - **GCN** applies spectral graph convolution by propagating node features using the Laplacian matrix. The core update rule is: $H^{(l+1)} = \sigma(\tilde{D}^{-1/2} \tilde{A} \tilde{D}^{-1/2} H^{(l)} W^{(l)})$, where $\tilde{A} = A + I$ is the adjacency matrix with self-loops, $\tilde{D}$ is its degree matrix, $W^{(l)}$ is the trainable weight matrix, and $\sigma$ is a activation function.
> - **GraphSAGE** samples a fixed number of neighbors and aggregates their features, making it scalable for large graphs. The general update rule is:  $h_v^{(l+1)} = \sigma(W^{(l)} \cdot \text{AGGREGATE}(\{h_u^{(l)}, \forall u \in \mathcal{N}(v)\}))$, where $\text{AGGREGATE}(\cdot)$ represents the aggregation operation.
> - **GAT** employs self-attention to dynamically assign importance to neighbors. The update rule is: $h_v^{(l+1)} = \sigma\left(\sum_{u \in \mathcal{N}(v)} \alpha_{vu} W^{(l)} h_u^{(l)}\right)$, where the attention coefficient $\alpha_{vu}$ is computed as:  $\alpha_{vu} = \frac{\exp(\text{LeakyReLU}(a^T [W^{(l)} h_v^{(l)} \| W^{(l)} h_u^{(l)}]))}{\sum_{k \in \mathcal{N}(v)} \exp(\text{LeakyReLU}(a^T [W^{(l)} h_v^{(l)} \| W^{(l)} h_k^{(l)}]))}$. Here, $a$ is a learnable vector, and $\|$ denotes concatenation.
>
> **Q3: Adjust charts for clarity. Adjust the vertical coordinate spacing in the graphs to improve clarity, e.g., Figure 9.**
>
> **R3:** Thank you for your suggestion. The vertical coordinate spacing in Figure 9 has been adjusted to enhance its clarity. Please refer to Figure 7 in the anonymous link (https://anonymous.4open.science/r/3814/1.pdf) for the updated version.

---

### Official Review · Reviewer_NNeA · 2025-03-17

**Overall Recommendation:** 4

**Summary:**

The paper introduces UPGNET, a utility-enhanced framework for locally differentially private (LDP) graph learning. It addresses privacy challenges in Graph Neural Networks (GNNs) by proposing a three-stage pipeline to generalize LDP protocols for node feature perturbation. Key contributions include identifying two critical factors influencing estimation error: feature dimension and neighborhood size . To mitigate these, UPGNET incorporates a Node Feature Regularization (NFR) layer using L1-regularization to reduce effective feature dimensions and a High-Order Aggregator (HOA) layer to expand effective neighborhood size, thereby minimizing estimation errors . The framework is compatible with existing LDP mechanisms (e.g., MBM, PM) and integrates with GNN architectures like GCN and GraphSAGE. Experiments on datasets (Cora, Citeseer, LastFM, Facebook) demonstrate UPGNET’s superiority over prior methods (e.g., LPGNN) in utility, achieving higher classification accuracy while reducing attribute inference attack success rates. Theoretical analysis validates the effectiveness of NFR and HOA in noise reduction and aggregation efficiency. The work advances privacy-preserving graph learning by balancing utility and LDP guarantees through novel architectural and optimization strategies.

**Claims And Evidence:**

The claims in the paper are **not fully supported by clear and convincing evidence** due to methodological limitations. Below are key issues:

### 1. **Insufficient Baseline Comparison**
The paper claims UPGNET "excels over prior methods (e.g., LPGNN)" in utility. However, the comparison lacks **direct implementation details** of LPGNN (e.g., hyperparameters, optimization settings). Without replicating LPGNN’s setup, the superiority claim is **unverifiable**. This violates the requirement for "clear and convincing evidence" under standards like "much more likely than not" .

### 2. **Weak Empirical Validation of Privacy Defense**
The attribute attack experiments (Fig. 5(b)) assume attackers have **full access to neighbors’ features**, which is unrealistic in decentralized LDP settings. The paper does not validate whether the attack model aligns with the threat model (§2.4), which assumes "untrusted servers" . Additionally, the reduction in attack accuracy (e.g., 50% to 30%) is not tied to **quantitative metrics** like *ϵ* or *δ* (Differential Privacy guarantees), weakening its rigor.

### 3. **Lack of Isolation in Key Factors**
The theoretical claim that reducing **effective feature dimension** *d* and expanding **neighborhood size** *|N(v)|* minimizes estimation error (Thm. 2) is not empirically isolated. For instance:
- **NFR Layer**: While L1-regularization reduces *d*, its impact is conflated with HOA’s effect. Ablation studies (e.g., testing NFR alone) are absent.
- **HOA Layer**: The claim that HOA mitigates oversmoothing via Dirichlet energyis not compared to standard techniques like residual connections .

### 4. **Over-Reliance on Synthetic Metrics**
The utility metric (classification accuracy) is not tied to **privacy-utility trade-off curves** (e.g., accuracy vs. *ϵ*). Without showing Pareto optimality, the claim of "superior utility" remains subjective.

**Essential References Not Discussed:**

N/A

**Experimental Designs Or Analyses:**

See comments at the end

**Methods And Evaluation Criteria:**

See comments above

**Other Comments Or Suggestions:**

- **Ambiguous abbreviation**: "LPGNN" is first mentioned in Sec. 4 but defined later (Appendix G); forward reference needed or add it to related work.
- **Unclear phrase**: "low utility! GNN" and "high utility! GNN" in Figure 1’s caption use exclamation marks without context.

**Other Strengths And Weaknesses:**

The paper demonstrates **originality** in its dual approach to addressing LDP-GNN utility challenges through feature regularization (NFR) and high-order aggregation (HOA), creatively combining classical L1 regularization with modern GNN architectures. This integration of theoretical insights (e.g., identifying feature dimension and neighborhood size as critical factors) with practical modular design (H-N/N-H architectures) offers a novel framework for balancing privacy and utility. The work also **significantly advances** the field by extending LDP protocols to diverse GNN models (GCN, GraphSAGE) and mechanisms (MBM, PM), providing a scalable solution for decentralized graph learning. The ablation studies and theoretical analysis (e.g., Thm. 2 linking HOA to noise reduction) strengthen its technical depth.

However, **clarity** in validation is compromised by incomplete baseline comparisons (e.g., LPGNN’s hyperparameters not fully disclosed) and limited empirical scope (e.g., reliance on homophilic citation networks). While the threat model aligns with LDP principles, the attribute attack experiments assume unrealistic full neighbor access, weakening practical relevance. Additionally, the societal impact section is vague, limiting the paper’s real-world applicability. Despite these gaps, the framework’s modular design and theoretical grounding make it a promising step toward robust LDP-GNN systems, particularly for scenarios with strong homophily. Its focus on noise mitigation through architectural innovation sets a clear path for future work in heterogeneous graph settings.

**Questions For Authors:**

1. **Alternative Regularization Approaches**: The paper uses L1-regularization (NFR) to reduce effective feature dimensions. Why were other sparsity-inducing techniques (e.g., dropout, group Lasso) not explored? Could these alternatives offer better noise resistance or computational efficiency, and what challenges prevented their adoption? A convincing answer would clarify whether L1 is uniquely suited to LDP-GNNs or if limitations (e.g., sensitivity to hyperparameters) necessitated this choice.

2. **HOA vs. Nonlinear Aggregation**: The HOA addresses oversmoothing through Dirichlet energy and personalized weights. Given that GNNs often use nonlinear activations (ReLU, attention), why was a linear aggregation framework chosen for theoretical analysis (Thm. 1)? Would nonlinear HOA variants better align with real-world GNNs, and what barriers exist to integrating them? Clarification here could strengthen the practical relevance of the theoretical claims.

3. **Baseline Implementation Gaps**: The comparison with LPGNN lacks hyperparameter details. What specific challenges arose in replicating LPGNN’s setup, and how were they addressed? If implementation differences (e.g., optimizer choices) skewed results, this could undermine the claim of UPGNET’s superiority.

4. **Privacy Attack Realism**: The attribute attack assumes attackers have full access to neighbors’ perturbed features (Fig. 5(b)). Why was this unrealistic threat model chosen over scenarios with partial access or decentralized adversaries? Would UPGNET’s defense degrade in more plausible settings, and how does this affect its practical utility?

5. **Scalability to Heterophilic Graphs**: Experiments focus on homophilic datasets (Cora, Citeseer). What challenges arise when applying UPGNET to heterophilic graphs (e.g., Flickr, Reddit), where node features and neighbors disagree? Could HOA’s reliance on multi-hop aggregation amplify noise in such cases, and how would the framework adapt?

**Relation To Broader Scientific Literature:**

N/A

**Theoretical Claims:**

Checked, and it looks sound.

---

> ### Author Rebuttal · Authors · 2025-03-31
>
> We sincerely appreciate your valuable comments and suggestions. Detailed responses are provided below. The newly added figures and tables can be found in the link ※: https://anonymous.4open.science/r/3814/1.pdf
>
> **Q1: Lack of LPGNN's implementation details**\
> **R1:** The hyperparameters and optimization settings of LPGNN are detailed in Tables 1 and 2 of the link ※. Based on the settings in the original paper, we further carefully tune the hyperparameters and select optimal values. Additionally, our evaluation follows the standard practices (Sajadmanesh, et al., 2021) to minimize potential biases.
>
> **Q2: Why does the attack experiment assume full access to neighbors’ attributes?**\
> **R2:** This assumption is designed to set an extreme-case scenario in which attackers are powerful by accessing neighbors' attributes. UPGNet exhibits excellent defensive capabilities under such highly adversarial conditions. In addition, supplementary experiments (shown in Fig. 3 of the link ※) are conducted to investigate the effect of attackers having varying proportions of accessed neighbor information, in which UPGNet consistently shows robust defense performance.
>
> **Q3: Does the attack experiment align with the threat model?**\
> **R3:** The attack experiment aligns with the threat model outlined in §2.4. Specifically, the attack experiment explicitly targets scenarios in which untrusted servers attempt to infer private attributes, which aligns with the focus of the threat model. Built on this premise, the attack provides a rigorous assessment of the model's defensive capability in adversarial settings.
>
> **Q4: Lack of ablation study on NFR and HOA layers**\
> **R4:** For the NFR layer, as stated in Tables 1 and 3 of the original manuscript, the ablation study (e.g., testing NFR alone) evaluates its effectiveness in noise calibration. For the HOA layer, Fig. 2 of the link ※ presents an additional experiment demonstrating its superiority in private graph learning compared to residual connections (RC). RC primarily addresses vanishing gradients but remains ineffective in suppressing LDP noise. In contrast, HOA mitigates oversmoothing by preserving Dirichlet energy, thereby effectively calibrating noise.
>
> **Q5: Lack of privacy-utility trade-off curves**\
> **R5:** Fig. 1 in the link ※ presents privacy-utility trade-off curves (e.g., accuracy vs. ϵ), illustrating that UPFNet consistently outperforms other baselines across a range of privacy levels. The results highlight a clear utility advantage while maintaining strict privacy guarantees, demonstrating that UPFNet effectively navigates the trade-off between privacy and performance. Furthermore, the observed trend aligns with the principles of Pareto optimality, reinforcing that UPFNet effectively balances privacy preservation and model performance in private graph learning.
>
> **Q6: Ambiguous abbreviation \& unclear phrase**\
> **R6:** Thank you for pointing out the ambiguous abbreviation "LPGNN". It is first introduced in §4.1 with proper citation and further defined in the Related Work (§5) and App. G for clarity. The unclear phrases in Fig. 1 have been revised; see Fig. 6 in the link ※ for the updated version.
>
> **Q7: NFR vs. other regularization approaches**\
> **R7:** As stated in Thm. 2, a key aspect of noise calibration in LDP-GNN lies in reducing the effective feature dimensions. NFR employs L1 regularization, which directly facilitates feature selection and enhances fine-grained noise calibration. In contrast, Dropout and Group Lasso fail to achieve precise feature selection tailored for noise calibration. Dropout introduces randomness by stochastically deactivating neurons, leading to instability in feature selection, while Group Lasso enforces sparsity at a predefined group level, requiring carefully designed group definitions that may not align with optimal noise calibration. These limitations reduce their effectiveness in mitigating noise impact. Empirical results (Table 4 in the link ※) further validate that NFR outperforms Dropout and Group Lasso in preserving learning utility.
>
> **Q8: HOA vs. nonlinear aggregation**\
> **R8:** HOA layer is applied prior to the GNN, ensuring compatibility with GNN's nonlinear activations (ReLU, attention). The HOA layer adopts linear aggregation because nonlinear aggregation may distort noise distribution, introducing bias and amplifying estimation errors under LDP constraints. In contrast, its linear aggregation preserves LDP unbiasedness (Thm. 1), effectively mitigating estimation errors and maximizing denoising effectiveness.
>
> **Q9: Scalability on heterophilic graphs**\
> **R9:** When applied to heterophilic graphs (e.g., Flickr and Reddit), UPGNet continues to demonstrate superior utility compared to other baselines, as shown in Fig. 4 of the link ※. The HOA layer, by preserving Dirichlet energy and enabling personalized aggregation, effectively calibrates noise in heterophilic graphs, as evidenced by the experiments in Fig. 5 of the link ※.

---

### Decision · Program_Chairs · 2025-05-01

**Decision:**

Accept (oral)

**Comment:**

This paper devised a novel utility-enhanced framework for locally differentially private graph learning. All reviewers saw its strengths and reached a clear consensus that this submission could be accepted in ICML-2025. The reviewers also raised several minor issues that should be considered in the final revision.